# Precedence-Constrained Winter Value for Effective Graph Data Valuation

**Hongliang Chi**[1]  **Wei Jin**[2]  **Charu Aggarwal**[3]  **Yao Ma**[1]

[1]Department of Computer Science, Rensselaer Polytechnic Institute, Troy, NY 12180
[2]Department of Computer Science, Emory University, Atlanta, GA 30322
[3]IBM T. J. Watson Research Center, Yorktown Heights, NY 10598
{chih3,may13}@rpi.edu, wei.jin@emory.edu, charu@us.ibm.com

## Abstract

Data valuation is essential for quantifying data's worth, aiding in assessing data quality and determining fair compensation. While existing data valuation methods have proven effective in evaluating the value of Euclidean data, they face limitations when applied to the increasingly popular graph-structured data. Particularly, graph data valuation introduces unique challenges, primarily stemming from the intricate dependencies among nodes and the growth in value estimation costs. To address the challenging problem of graph data valuation, we put forth an innovative solution, **P**recedence-**C**onstrained **Winter** (`PC-Winter`) Value, to account for the complex graph structure. Furthermore, we develop a variety of strategies to address the computational challenges and enable efficient approximation of `PC-Winter`. Extensive experiments demonstrate the effectiveness of `PC-Winter` across diverse datasets and tasks.

## 1 Introduction

The abundance of training data has been a key driver of recent advancements in machine learning (ML) [51]. As models and the requisite training data continue to expand in scale, data valuation has gained significant attention due to its ability to quantify the usefulness of data for ML tasks and determine fair compensation [28, 34]. Notable techniques in this field include Data Shapley [13] and its successors [20, 39, 29], which have gained prominence in assessing data value. Despite the promise of these methods, they are primarily designed for Euclidean data, where samples are often assumed to be independent and identically distributed (i.i.d.). Given the prevalence of graph-structured data in the real world [10, 31, 22], there arises a compelling need to perform data valuation for graphs. However, due to the interconnected nature of samples (nodes) on graphs, existing data valuation frameworks are not directly applicable to addressing the graph data valuation problem.

In particular, designing data valuation methods for graph-structured data faces several fundamental challenges: **Challenge I:** Graph machine learning algorithms such as Graph Neural Networks (GNNs) [19, 37, 41] often involve both labeled and unlabeled nodes in their model training process. Therefore, unlabeled nodes, despite their absence of explicit labels, also hold intrinsic value. Existing data valuation methods, which typically assess a data point's value based on its features and the associated label, do not readily accommodate the valuation of unlabeled nodes within graphs. **Challenge II:** Nodes in a graph contribute to model performance in an interdependent and complex way: (1) Unlabeled nodes, while not providing direct supervision, can contribute to model performance by potentially affecting multiple labeled nodes through message-passing. (2) Labeled nodes, on the other

Submitted to the 38th Conference on Neural Information Processing Systems (NeurIPS 2024) Track on Datasets and Benchmarks. Do not distribute.

hand, contribute by providing direct supervision signals for model training, and similarly to unlabeled nodes, they also contribute by affecting other labeled nodes through message-passing. **Challenge III:** Traditional data valuation methods are often computationally expensive due to repeated retraining of models [13]. The challenge is magnified in the context of graph-structured data, where samples contribute to model performance in multifaceted manners. Additionally, the inherent message-passing mechanism in GNN models further amplifies the computational demands for model re-training.

In this work, we make *the first attempt* to explore the challenging graph data valuation problem, to the best of our knowledge. In light of the aforementioned challenges, we propose the **P**recedence-**C**onstrained **Winter** (`PC-Winter`) Value, a pioneering approach designed to intricately unravel and analyze the contributions of nodes within graph structures, thereby offering a detailed perspective on the valuation of graph elements. Our key contributions are as follows:

- We formulate the graph data valuation problem as a unique cooperative game [38] with special coalition structures. Specifically, we decompose each node in the graph into several "players" within the game, each representing a distinct contribution to model performance. We then devise the `PC-Winter` to address the game, enabling the accurate valuation of all players. The `PC-Winter` values of these players can be conveniently combined to generate values for nodes and edges.
- To tackle the computational challenges of calculating `PC-Winter` values, we develop a set of strategies including *hierarchical truncation* and *local propagation*. These strategies together enable an efficient approximation of `PC-Winter` values.
- Extensive experiments on various datasets and tasks, along with detailed ablation studies and parameter analyses, validate the effectiveness of `PC-Winter` and provide insights into its behavior.

## 2 Preliminary and Related Work

In this section, we delve into some fundamental concepts that are essential for developing our methodology. More extensive literature exploration can be found in Appendix A.

### 2.1 Cooperative Game Theory

Cooperative game theory explores the dynamics where players, or decision-makers, can form alliances, known as coalitions, to achieve collectively beneficial outcomes [2, 7]. The critical components of such a game include a *player set* $\mathcal{P}$ consisting of all players in the game and a *utility function* $U(\cdot)$, which quantifies the value or payoff that each coalition of players can attain. Shapley Value [32] is developed to fairly and efficiently distribute payoffs (values) among players.

**Shapley value.** The Shapley value $\phi_i(\mathcal{P}, U)$ for a player $i \in \mathcal{P}$ can be defined on permutations of $\mathcal{P}$ as follows.

$$\phi_i(\mathcal{P}, U) = \frac{1}{|\Pi(\mathcal{P})|} \sum_{\pi \in \Pi(\mathcal{P})} [U(\mathcal{P}_i^\pi \cup \{i\}) - U(\mathcal{P}_i^\pi)] \tag{1}$$

where $\Pi(\mathcal{P})$ denotes the set of all possible permutations of $\mathcal{P}$ with $|\Pi(\mathcal{P})|$ denoting its cardinality, and $\mathcal{P}_i^\pi$ is predecessor set of $i$, i.e, the set of players that appear before player $i$ in a permutation $\pi$:

$$\mathcal{P}_i^\pi = \{j \in \mathcal{P} \mid \pi(j) < \pi(i)\}. \tag{2}$$

The Shapley value considers each player's contribution to every possible coalition they could be a part of. Specifically, in Eq. (1), for each permutation $\pi$, the *marginal contribution* of player $i$ is calculated as the difference in the utility function $U$ when player $i$ is added to an existing coalition $\mathcal{P}_i^\pi$. The Shapley value $\phi_i(\mathcal{P}, U)$ for $i$ is the average of these marginal contributions across all permutations in $\Pi(\mathcal{P})$. The Shapley value has been widely applied in ML for various tasks such as data valuation [13, 17] and model explanation [24, 11]. In the context of graph ML, it has been primarily used for GNN explainability [8, 47, 25, 1]. A more detailed discussion on Shapley Value on graph ML can be found in Appendix A.3.

**Winter Value.** The Shapley value is to address cooperative games, where players collaborate freely and contribute on an equal footing. However, in many practical cases, cooperative games, exhibit a *Level Coalition Structure* [26, 36, 48], reflecting a hierarchical organization. For instance, consider a corporate setting where different tiers of management and staff contribute to a project in

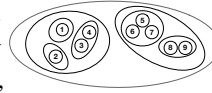

Figure 1: Level Coalition Structure

varying capacities and with differing degrees of decision-making authority. Players within such a game are hierarchically categorized into nested coalitions with several levels, as depicted in Figure 1. The outermost and largest ellipse represents the entire coalition and each of the smaller ellipse within the largest ellipse symbolizes a "sub-coaliation" at various hierarchical levels. Collaborations originate within the smallest sub-coalitions at the base level (illustrated by the innermost ellipses in Figure 1. These base units are then integrated into the next level, facilitating inter-coalition collaboration and enabling contributions to ascend to higher levels. This bottom-up flow of contributions continues, with each layer consolidating and passing on inputs to the next, culminating in a multi-leveled collaborative contribution to the final objective of the entire coalition. To accommodate such complex Level Coalition Structure, Winter value [40] was introduced. Winter value follows a similar permutation-based definition as Shapley Value (Eq. (1)) but with only a specific subset of permutations that respect the Level Coalition Structure. In these permutations, members of the same sub-coalition, regardless of the level, must appear in an unbroken sequence without interruptions. This ensures that the value attributed to each player is consistent with the level structure of the coalition. A formal definition of the Winter value can be found in Appendix B.

## 2.2  Data Valuation and Data Shapley

Data valuation quantifies the contribution of data points for machine learning tasks. The seminal work [13] introduces Data Shapley, applying cooperative game theory to data valuation, where training samples are the *players* $\mathcal{P}$, and the *utility function* $U$ assesses a model's performance on subsets of these players using a validation set. With $\mathcal{P}$ and $U$, data values can be calculated with Eq. (1). However, Data Shapley and subsequent methods [13, 20, 39] primarily focus on i.i.d. data, overlooking potential coalitions or dependencies among data points.

## 2.3  Graphs and Graph Neural Networks

Consider a graph $\mathcal{G} = \{\mathcal{V}, \mathcal{E}\}$ where $\mathcal{V}$ denotes the set of nodes and $\mathcal{E}$ denotes the set of edges. Each node $v_i \in \mathcal{V}$ carries a feature vector $\mathbf{x}_i \in \mathbb{R}^d$, where $d$ is the dimensionality of the feature space. Additionally, each node $v_i$ is associated with a label $y_i$ from a set of possible labels $\mathcal{C}$. We assume that only a subset $\mathcal{V}_l \subset \mathcal{V}$ are with known labels.

GNNs [19, 37, 41] are prominent models for graph ML tasks. Specifically, from a local perspective for node $v_i$, the $k$-th GNN layer generally performs a feature averaging process as $\mathbf{h}_i^{(k)} = \frac{1}{deg(v_i)} \sum_{v_j \in \mathcal{N}(v_i)} \mathbf{W} \mathbf{h}_j^{(k-1)}$, where $\mathbf{W}$ is the parameter matrix, $deg(v_i)$ and $\mathcal{N}(v_i)$ denote the degree and neighbors of node $v_i$, respectively. After a total of $K$ layers, $\mathbf{h}_i^{(K)}$ are utilized as the learned representation of $v_i$. Such a feature aggregation process can be also described with a $K$-level *computation tree* [15] rooted on node $v_i$.

**Definition 1** (Computation Tree). *For a node $v_i \in \mathcal{V}$, its $K$-level computation tree corresponding to a $K$-layer GNN model is denoted as $\mathcal{T}_i^K$ with $v_i$ as its root node. The first level of the tree consists of the immediate neighbors of $v_i$, and each subsequent level is formed by the neighbors of nodes in the level directly above. This pattern of branching out continues, expanding through successive levels of neighboring nodes until the depth of the tree grows to $K$.*

The feature aggregation process in a $K$-layer GNN can be regarded as a bottom-up feature propagation process in the computation tree, where nodes in the lowest level are associated with their initial features. Therefore, the final representation $\mathbf{h}_i^{(K)}$ of a node $v_i$ is affected by all nodes within its $K$-hop neighborhood, which is referred to as the *receptive field* of node $v_i$. The GNN model is trained using the $(\mathbf{h}_i^{(K)}, y_i)$ pairs, where each labeled node $v_i$ in $\mathcal{V}_l$ is represented by its final representation and corresponding label. *Thus, in addition to labeled nodes, those unlabeled nodes that are within the receptive field of labeled nodes also contribute to model performance.*

## 3  Methodology

In classic machine learning models designed for Euclidean data, such as images and texts, training samples are typically assumed as i.i.d. Thus, each labeled sample contributes to the model performance by directly providing supervision signals through the training objective. However, due to the

interdependent nature of graph data, nodes in a graph contribute to GNN performance in a more complicated way, which poses unique challenges. Specifically, as discussed in Section 2.3, both labeled and unlabeled nodes are involved in the training stage through the feature aggregation. Next, we discuss how these nodes contribute to GNN performance.

**Observation 1.** *Unlabeled nodes influence GNN performance by affecting the final representation of labeled nodes. On the other hand, labeled nodes can contribute to GNN performance in two ways: (1) they provide direct supervision signals to GNN with their labels, and (2) just like unlabeled nodes, they can impact the final representation of other labeled nodes through feature aggregation. Note that both labeled nodes and unlabeled nodes can affect the final representations of multiple labeled nodes, as long as they lie within the receptive field of these labeled nodes. Hence, a single node can make multifaceted and heterogeneous contributions to GNN performance by affecting multiple labeled nodes in various manners.*

### 3.1 The Graph Data Valuation Problem

Based on Observation 1, due to the heterogeneous and diverse effects of labeled and unlabeled nodes, it is necessary to perform fine-grained data valuation on graph data elements. In particular, we propose to decompose a node into distinct "duplicates" corresponding to their impact on different labeled nodes. We then aim to obtain values for all "duplicates" of these nodes. This could clearly express and separate how nodes impact GNN performance in various aspects. Following existing literature [13, 39, 43], we approach the graph data value problem through a cooperative game. Next, we introduce the *player set* and the *utility function* of this game. In general, we define the graph data valuation game based on $K$-layer GNN models.

**Definition 2** (Player Set). *The player set $\mathcal{P}$ in a graph data valuation game is defined as the union of nodes in the computation trees of labeled nodes. Duplication of nodes may occur within a single computation tree $\mathcal{T}_i^K$ or across different labeled nodes' computation trees. In the graph data valuation game, these potential duplicates are treated as distinct players, uniquely identified by their paths to the corresponding labeled node. We define the player set $\mathcal{P}$ as the set of all these distinct players across the computation trees of all labeled nodes in $\mathcal{V}_l$.*

**Definition 3** (Utility Function). *Given a subset $\mathcal{S} \subset \mathcal{P}$, we first generate a node-induced graph $G_{in}(\mathcal{S})$ using their corresponding edges in the computation trees. Then, a GNN model $\mathcal{A}$ is trained on the induced graph $G_{in}(\mathcal{S})$. Its performance is evaluated on a held-out validation set to serve as the utility of $\mathcal{S}$, calculated as $U(\mathcal{S}) = acc(\mathcal{A}(G_{in}(\mathcal{S})))$, where acc measures the accuracy of the trained GNN model $\mathcal{A}(G_{in}(\mathcal{S}))$ on a held-out validation set.*

The goal of the graph data valuation problem is to assign a value to all players in $\mathcal{P}$ with the help of the *utility function* $U$. When calculated properly, these values are supposed to provide a detailed understanding of how players in $\mathcal{P}$ contribute to the GNN performance in a fine-grained manner. Furthermore, these values can be flexibly combined to generate higher-level values for nodes and edges, which will be discussed in Section 3.5.

### 3.2 Precedence-Constrained Winter Value

As discussed in Section 2.3, the final representations of a labeled node $v_i$ come from the hierarchical collaboration of all players in the computation tree $\mathcal{T}_i^K$. These labeled nodes with the updated representations then contribute to the GNN performance through the training objective. Such a contribution process forms a hierarchical collaboration between the players in $\mathcal{P}$, which can be illustrated with a *contribution tree* $\mathcal{T}$ as shown in Figure 2a. In particular, the *contribution tree* $\mathcal{T}$ is constructed by linking the root nodes of the computation trees of all labeled nodes with a dummy node representing the GNN training objective $\mathcal{O}$. In Figure 2a, for the ease of illustration, we set $K = 2$, include only 2 labeled nodes, i.e, $v_0, v_1$, and utilize $w_i, u_i$ to denote the nodes in the lower level. The subtree rooted at a labeled node $v_i \in \mathcal{V}$ is the corresponding computation tree $\mathcal{T}_i^2$. With this, we observe the following about the coalition structure of the graph data valuation game.

**Observation 2** (Level Coalition Structure). *As shown in Figure 2a, the players in $\mathcal{P}$ hierarchically collaborate to contribute. At the bottom level, the players are naturally grouped by their parents. Specifically, players with a common parent such as $u_0, u_1, u_2$ with their parent $w_0$, establish a foundation sub-coalition. This sub-coalition is clearly depicted in Figure 2b. Moving up the tree,*

these parent nodes, like $w_0$, serve as "delegates" for their respective sub-coalitions, further engaging in collaborations with other sub-coalitions. This interaction forms higher-level sub-coalitions, such as the one between $w_0$, $w_1$, $w_2$, and $v_0$ in Figure 2b, indicating inter-coalition cooperation. This ascending process of coalition formation continues until the root node $\mathcal{O}$ is reached, which represents the objective of the entire coalition consisting of all players. The depicted hierarchical collaboration process aligns with the Level Coalition Structure discussed in Section 2.1.

While the contribution tree shares similarities with the Level Coalition Structure illustrated in Section 2.1, a pivotal distinction lies in the representation and function of "delegates" (highlighted in red in Figure 2b) within each coalition. In the traditional Level Coalition Structure, contributions within a sub-coalition are made collectively, with each player or lower-level sub-coalition participating on an equitable basis. In contrast, the contribution tree framework distinguishes itself by designating a "delegate" within each sub-coalition, a player that represents and advances the collective contributions, establishing a directed and tiered flow of influence, hence forming a Unilateral Dependency Structure.

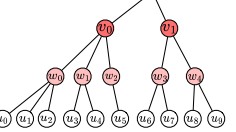 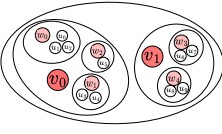

(a) Contribution Tree    (b) Coalition Structure

Figure 2: Graph Data Valuation Game Structure

**Observation 3** (Unilateral Dependency Structure). *In the contribution tree framework, a player $p \in \mathcal{P}$ contributes to the final objective through a hierarchical pathway facilitated by its ancestors (its "delegates" at different levels). Therefore, the collaboration between players in $\mathcal{P}$ exhibits a Unilateral Dependency Structure, where a player $p$'s contribution is dependent on its ancestors.*

According to these two observations, the players demonstrate unique coalition structures in the graph data valuation game. We aim to propose a permutation-based valuation framework similar to Eq. (1) to address the cooperative game with both Level Coalition Structure and Unilateral Dependency Structure. In particular, instead of utilizing all the permutations as in Eq. (1), only the *permissible permutations* aligning with such coalition structures are included in the value calculations. As we described in Section 2.1, cooperative games with Level Coalition Structure have been addressed by the Winter value [40, 4]. Specifically, a permutation respecting the Level Coalition Structure must ensure that players in the same (sub-)coalition, regardless of its level, are grouped together without interruption from other players [40]. In our scenario, any subtree of the contribution tree corresponds to a sub-coalition as demonstrated in Figure 2. Hence, we need to ensure that for any player $p \in \mathcal{P}$, the player $p$ and its descendants in the contribution tree should be grouped together in the permutation. For example, the players $w_0, u_0, u_1, u_2$ should present together as a group in the permutation with potentially different orders. On the other hand, to ensure the Unilateral Dependency Structure, a permutation must maintain a partial order. Specifically, for any player $p$ in the permutation, its descendants must present in later positions in the permutation than $p$. Otherwise, the descendants of $p$ cannot make non-trivial contributions, resulting in 0 marginal contributions.

We formally define the *permissible permutations* that align with both Level Coalition Structure and Unilateral Dependency Structure utilizing the following two constraints.

**Constraint 1** (Level Constraint). *For any player $p \in \mathcal{P}$, the set of its descendants in the contribution tree is denoted as $\mathcal{D}(p)$. Then, a permutation $\pi$ aligning with the Level Coalition Structure satisfies the following Level Constraint: $|\pi[i] - \pi[j]| \leq |\mathcal{D}(p)|, \forall i, j \in \mathcal{D}(p) \cup p, \forall p \in \mathcal{P}$, where $\pi[i]$ denotes the positional rank of the $i$ in $\pi$.*

**Constraint 2** (Precedence Constraint). *A permutation $\pi$ aligning with the Unilateral Dependency Structure satisfies the following Precedence Constraint: $\pi[p] < \pi[i], \forall i \in \mathcal{D}(p), \forall p \in \mathcal{P}$.*

We denote the set of *permissible permutations* satisfying both *Level Constraint* and *Precedence Constraint* as $\Omega$. Then, we define the **P**recedence-**C**onstrained **Winter** (PC-Winter) value for a player $p \in \mathcal{P}$ with the permutations in $\Omega$ as follows.

$$\psi_p(\mathcal{P}, U) = \frac{1}{|\Omega|} \sum_{\pi \in \Omega} \left( U\left(\mathcal{P}_p^\pi \cup p\right) - U\left(\mathcal{P}_p^\pi\right) \right), \tag{3}$$

where $U(\cdot)$ is the utility function (see Definition 3), and $\mathcal{P}_p^\pi$ denotes the predecessor set of $p$ in $\pi$ as defined in Eq. (2).

### 3.3 Permissible Permutations for `PC-Winter`

To calculate `PC-Winter` value, it is required to obtain all permissible permutations. A straightforward way is to enumerate all permutations and only retain the permissible permutations. However, such an approach is extremely computationally intensive and typically not feasible in reality. In this section, to address this challenge, we propose a novel method to directly generate these permutations by traversing the contribution tree with Depth-First Search (DFS). Specifically, each DFS traversal results in a *preordering*, which is a list of the nodes (players) in the order that they were visited by DFS. Such a *preordering* naturally defines a permutation of $\mathcal{P}$ by simply removing the dummy node in the contribution tree from the *preordering*. By iterating all possible DFS traversals of the contribution tree, we can obtain all permutations in $\Omega$, which is demonstrated in the following theorems.

**Theorem 1** (Specificity). *Given a contribution tree $\mathcal{T}$ with a set of players $\mathcal{P}$, any DFS traversal over the $\mathcal{T}$ results in a permissible permutation of $\mathcal{P}$ that satisfies both the Level Constraint and Precedence Constraint.*

**Theorem 2** (Exhaustiveness). *Given a contribution tree $\mathcal{T}$ with a set of players $\mathcal{P}$, any permissible permutation $\pi \in \Omega$ can be generated by a corresponding DFS traversal of $\mathcal{T}$.*

The proofs for two theorems can be found in Appendix C. Theorem 1 demonstrates that DFS traversals *specifically* generate *permissible permutations*. On the other hand, Theorem 2 ensures the *exhaustiveness* of generation, which allows us to obtain all permutations in $\Omega$ by DFS traversal. Together, these two theorems ensure us to *exactly* generate the set of *permissible permutations* $\Omega$.

Notably, the calculation of `PC-Winter` value involves two steps: 1) generating $\Omega$ with DFS traversals; and 2) calculating the `PC-Winter` value according to Eq. (3). Nonetheless, it can be done in a streaming way while we perform the DFS traversals. Specifically, once we reach a player $p$ in a DFS traversal, we can immediately calculate its marginal contribution. The `PC-Winter` values for all players are computed by averaging their marginal contributions from all possible DFS traversals.

### 3.4 Efficient Approximation of `PC-Winter`

Calculating the `PC-Winter` value for players in $\mathcal{P}$ is infeasible due to computational intensity, arising from: 1) The exponential growth in the number of permissible permutations with more players, rendering exhaustive enumeration intractable; 2) The necessity to re-train the GNN within the utility function for each permutation, a process repeated $|\mathcal{P}|$ times to account for every player's marginal contribution; and 3) The intensive computation involved in GNN re-training, requiring feature aggregation over the graph that increases in complexity with the graph's size. These challenges necessitate an efficient approximation method for `PC-Winter` valuation in practical applications. We propose three strategies to address these computational issues.

**Permutation Sampling.** Following Data Shapley [13], we adopt Monte Carlo (MC) sampling to randomly sample a subset of permissible permutations denoted as $\Omega_s$. Then, we utilize $\Omega_s$ to replace $\Omega$ in Eq. (3) for approximating `PC-Winter` value.

**Hierarchical Truncation.** GNN models often demonstrate a phenomenon of *neighborhood satura-tion*, i.e, these models achieve satisfactory performance even when trained on a subgraph using only a small subset of neighbors, rather than the full neighborhood [14, 23, 45, 5], indicating diminishing returns from additional neighbors beyond a certain point. This indicates that for a player $p$ in a permissible permutation $\pi$ generated by DFS over the contribution tree, the marginal contributions of its late visited child players are insignificant. Thus, we propose hierarchical truncation for efficiently obtaining the marginal contributions by directly approximating insignificant values as $0$. Specifically, during the DFS traversal, given a truncation ratio $r$, we only compute actual marginal contributions for players in the first $1 - r$ portion of each node's child subtrees, approximating the marginal contributions of players in the remaining subtrees as $0$. For example, in Figure 2a, given a truncation ratio $r = 2/3$, when DFS reaches player $v_0$, we only calculate marginal contributions for players in the subtree rooted at $w_0$. Furthermore, in the subtree rooted at $w_0$, due to the hierarchical truncation, only the marginal contribution of $u_0$ is evaluated, those for node $u_1$ and $u_2$ are set to $0$. This approach is further optimized by adjusting truncation ratios based on the tree level, accommodating varying contribution patterns across levels. In particular, we organize the pair of truncation ratio as $r_1$-$r_2$,

indicating we truncate $r_1$ (or $r_2$) portion of subtrees (or child players) of $v_i$ (or $w_i$). We show how the hierarchical truncation helps tremendously reduce the model re-training in Appendix D.

**Local Propagation.** To enhance scalability, we leverage SGC [41] in our utility function, which simplifies GNNs by aggregating node features before applying an MLP. According to the Level Constraint (Constraint 1), the players within the same computation tree are grouped together in the permutation. Therefore, the induced graph of any coalition $\mathcal{P}_p^\pi$ defined by a permissible permutation consists of a set of separated computation trees (or a partial computation tree corresponding to the last visited labeled node in $\mathcal{P}_p^\pi$). A key observation is that the feature aggregation process for the labeled nodes can be done independently within their own computation trees. Hence, instead of performing the feature propagation for the entire induced graph, we propose to perform *local propagation* only on necessary computation trees. In particular, the aggregated representation for a labeled node is fixed after we traverse its entire computation tree in DFS. Therefore, for evaluating a player $p$'s marginal contribution, only the partial computation tree of the last visited labeled node requires *local propagation*, minimizing feature propagation efforts.

The `PC-Winter` values for all players are approximated with these three strategies in a streaming manner. In particular, we randomly traverse the contribution tree with DFS for $|\Omega_s|$ times. During each DFS traversal, the marginal contributions for all players in $\mathcal{P}$ are efficiently obtained with the help of *hierarchical truncation* and *local propagation*. The marginal contributions calculated through these $|\Omega_s|$ DFS traversals are averaged to approximate the `PC-Winter` value for all players. In Appendix H.5, we provide a detailed complexity analysis of the `PC-Winter` algorithm.

### 3.5 From `PC-Winter` to Node and Edge Values

The `PC-Winter` values for players in $\mathcal{P}$ can be flexibly combined to obtain the values for elements in the original graph, which are illustrated in this section. Specifically, as discussed in Section 3.1, multiple "duplicates" of a node $v \in \mathcal{V}$ in the original graph may potentially present in $\mathcal{P}$. Thus, we could obtain *node value* for the node $v$ by summing the `PC-Winter` values of all its "duplicates" in $\mathcal{P}$. On the other hand, each player (except for the rooted labeled players) in $\mathcal{P}$ corresponds to an "edge" in the contribution tree as identified by the player and its parent. For instance, in Figure 2a, the player $u_0$ corresponds to "edge" connecting $u_0$ and $w_0$. Therefore, DFS traversals also generate permutations for these "edges". From this perspective, the marginal contribution for a player $p$ calculated through a DFS traversal can be also regarded as the marginal contribution of its corresponding edge, if we treat this process as gradually adding "edges" to connecting the players in $\mathcal{P}$. Hence, the `PC-Winter` values for players in $\mathcal{P}$ can be regarded as `PC-Winter` values for their corresponding "edges" in the contribution tree. Multiple "duplicates" of an edge $e \in \mathcal{E}$ in the original graph may be present in the contribution tree. Hence, similar to the *node values*, we define the *edge value* for $e \in \mathcal{E}$ by taking the summation of the `PC-Winter` value for all its "duplicates" in the contribution tree.

## 4 Experiment

**Datasets and Settings.** We assess the proposed approach on six real-world benchmark datasets: Cora, Citeseer, and Pubmed [30], Amazon-Photo, Amazon-Computer, and Coauther-Physics [33]. The detailed statistics of datasets are summarized in Table 2 in Appendix G. Our experiments focus on the inductive node classification task. The detailed setup of the inductive setting can be found in Appendix G.1. To obtain the `PC-Winter` values, we run permutations in a streaming way as described in Section 3.4. This process terminates with a convergence criterion as detailed in Appendix G.4. `PC-Winter` typically terminates with a different number of permutations for different datasets. The other hyper-parameters are detailed in Appendix G.5.

### 4.1 Dropping High-Value Nodes

In this section, we aim to evaluate the quality of data values produced by `PC-Winter` via dropping high-value nodes from the graph. Dropping high-value nodes is expected to significantly diminish performance, and thus the performance observed after removing high-value nodes serves as a strong indicator of the efficacy of graph data valuation. Notably, `PC-Winter` values values are calculated as described in Section 3.5.

To demonstrate the effectiveness of `PC-Winter`, we include `Random` value, `Degree` value, Leave-one-out (`LOO`) value, and `Data Shapley` value as baselines. A more detailed description of these baselines is included in Appendix G.6. To conduct node-dropping experiments, nodes are ranked by their assessed values for each method and removed sequentially from the training graph $\mathcal{G}_{tr}$. After each removal, we train a GNN model based on the remaining graph and evaluate its performance on the testing graph $\mathcal{G}_{te}$. Performance changes are depicted through a curve that tracks the model's accuracy as nodes are progressively eliminated. Labeled nodes often contribute more significantly to model performance than unlabeled nodes because they directly offer supervision. Thereby, with accurately assigned node values, labeled nodes should be prioritized for removal over unlabeled nodes. We empirically validate this hypothesis in Figure 6, discussed in Appendix E. Specifically, in nearly all datasets, our observations reveal that the majority of labeled nodes are removed prior to the unlabeled nodes by both `PC-Winter` and `Data Shapley`. This leads to a plateau in the latter portion of the performance curves since a GNN model cannot be effectively trained with only unlabeled nodes. Consequently, this scenario significantly hampers the ability to assess the value of unlabeled nodes. Therefore, we propose to conduct separate assessments for the values of labeled and unlabeled nodes. Here, we only inlcude the results for unlabeled nodes, while the results for labeled nodes are presented in Appendix F.

**Results and Analysis.** Figure 3 illustrates the performance comparison between `PC-Winter` and other baselines across various datasets. From Figure 3, we make the following observations. First, the removal of high-value unlabeled nodes identified by `PC-Winter` consistently results in the most significant decline in model performance across various datasets. This is particularly evident after removing a relatively small fraction (10%-20%) of the highest-value nodes. This trend underscores the importance of high-value nodes. Notably, in most datasets `PC-Winter` outperforms the best baseline method, `Data Shapley`, by a considerable margin, highlighting its effectiveness. Second, the decrease in performance caused by our method is not only substantial but also persistent throughout the node-dropping process, further validating the effectiveness of

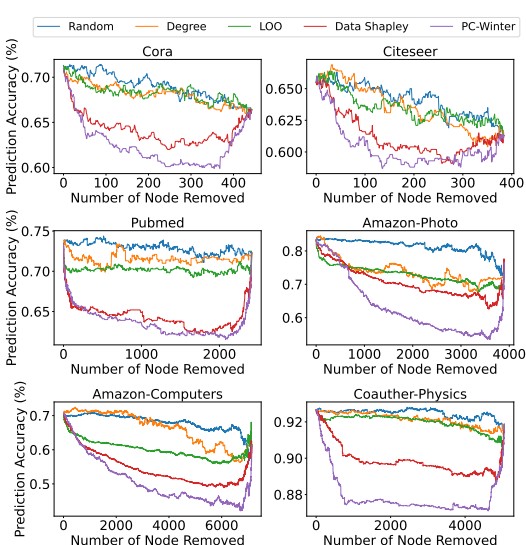

Figure 3: Dropping High-Value Nodes

`PC-Winter`. Third, the performance curves of `PC-Winter` and `Data Shapley` eventually rebound towards the end. This rebound corresponds to the removal of unlabeled nodes that make negative contributions. Their removal aids in improving performance, ultimately reaching the MLP performance when all nodes are excluded. This upswing not only evidences the discernment of `PC-Winter` and `Data Shapley` in ascertaining node values but also showcases the particularly acute precision of `PC-Winter`. These insights collectively affirm the capability of `PC-Winter` in accurately assessing node values.

### 4.2 Adding High-Value Edges

In this section, we explore the impact of adding high-value elements to a graph, providing an alternative perspective to validate the effectiveness of data valuation. Notably, adding high-value nodes to a graph typically involves the concurrent addition of edges, which complicates the addition process. Thus, we target the addition of high-value edges, providing a complementary perspective to our analysis. As described in Section 3.5, the flexibility of `PC-Winter` allows for obtaining edge values without a separate "reevaluation" process for edges.

Here, we keep all nodes in $\mathcal{G}_{tr}$ and sequentially add edges according to the edge values in descending order, starting with the highest-valued ones. Similar to the node-dropping experiments, the effectiveness of the edge addition is shown through performance curves. We include `Random` value,

Edge-Betweeness, Leave-one-out (LOO) as baselines. Notably, here, Random and LOO specifically pertain to edges, and while we use the same terminology as in the prior section, they are distinct methods, which are detailed in Appendix G.6.

**Results and Analysis.** Figure 4 illustrates that the Random, LOO, and Edge-Betweeness baselines achieve only linear performance improvements with the addition of more edges, failing to discern the most impactful ones for a sparse yet informative graph. In contrast, the inclusion of edges based on the PC-Winter value results in a steep performance climb, affirming the PC-Winter's efficacy in pinpointing key edges. Notably, the Cora dataset reaches full-graph performance using merely 8% of the edges selected by PC-Winter. Moreover, with just 10% of PC-Winter-selected edges, the accuracy climbs to 72.9%, outperforming the full graph's 71.3%, underscoring PC-Winter's capability to identify valuable edges. This trend is generally consistent across other datasets as well.

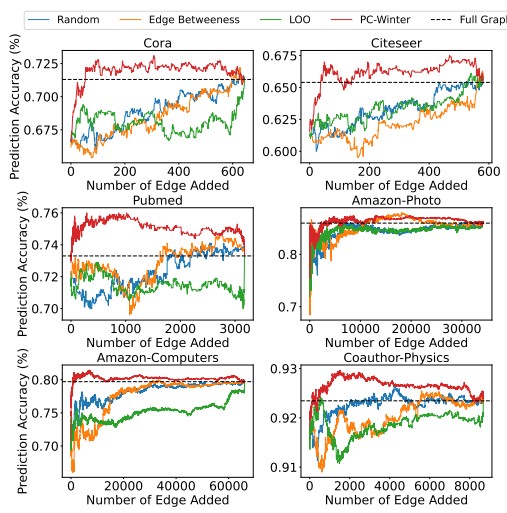

Figure 4: Adding the High-Value Edges

### 4.3 Ablation Study, Parameter and Efficiency Analysis

In this section, we conduct an ablation study, parameter analysis, and efficiency analysis to gain deeper insights into PC-Winter using node-dropping experiments.

**Ablation Study.** We conduct an ablation study to understand how the two constraints in Section 3.2 affect the effectiveness of PC-Winter. We introduce two variants of PC-Winter by lifting one of the constraints for the permutations. In particular, we define PC-Winter-L using the permutations satisfying the Level Constraint. Similarly, PC-Winter-P is defined with permutations only satisfying Precedence Constraint. As shown in Figure 5, PC-Winter value outperforms the PC-Winter-L and PC-Winter-P on both datasets, which demonstrates that both constraints are crucial for PC-Winter. Additional results on other datasets are provided in Appendix H.1.

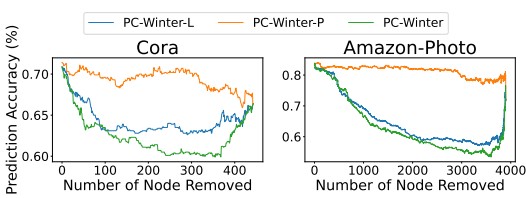

Figure 5: Ablation Study

**Parameter Analysis.** We conduct parameter analyses to investigate the impact of permutation number and truncation ratios on PC-Winter's performance. The results reveal that PC-Winter achieves robust performance even with a significantly reduced number of permutations and high truncation ratios. Detailed findings are presented in Appendix H.2 and Appendix H.3, respectively.

**Efficiency Analysis.** We compare the efficiency of PC-Winter and Data Shapley. Analysis of converged permutation count and time per permutation across 6 datasets underscores PC-Winter's significantly higher efficiency. A comprehensive breakdown is available in Appendix H.4.

## 5 Conclusion

In this paper, we introduce PC-Winter, an innovative approach for effective graph data valuation. The method is specifically designed for graph-structured data and addresses the challenges posed by unlabeled elements and complex node dependencies within graphs. Furthermore, we introduce a set of strategies for reducing the computational cost, enabling efficient approximation of PC-Winter. Extensive experiments demonstrate the practicality and effectiveness of PC-Winter in various datasets and tasks. While PC-Winter demonstrates improved efficiency compared to Data Shapley, we acknowledge that further efficiency enhancements are crucial to fully unlock the potential of graph data valuation in real-world applications. Our work can be seen as a foundation for future research in this direction.

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

# A   Additional Related Work

This section presents an extended review of related works, offering a broader and more nuanced exploration of the literature surrounding Data Valuation and Graph Neural Networks.

## A.1   Data Valuation

Data Shapley is proposed in [13] which computes data values with Shapley values in cooperative game theory. Beta Shapley [20] is a further generalization of Data Shapley by relaxing the efficiency axiom of the Shapley value. Data Banzhaf [39] offers a data valuation method which is robust to data noises. Data Valuation with Reinforcement Learning is also explored by [46]. KNN-Shapley [17] estimates the shapley Value for the K-Nearest Neighbours algorithm in linear time. CS-Shapley [29] provides a new valuation method that differentiate in-class contribution and out-class contribution. Data-OOB [21] proposes a data valuation method for a bagging model which leverages the out-of-bag estimate. Just, Hoang Anh, et al [18] introduce a learning-agnostic data valuation framework by approximating the utility of a dataset according to its class-wise Wasserstein distance. Another training-free data valuation method utilizing the complexity-gap score is proposed at the same time [27]. However, those methods are not designed for the evaluation of data value of graph data which bears higher complexity due to the interconnections of individual nodes.

## A.2   Graph Neural Networks

Graph Neural Networks (GNNs) generate informative representations from graph-structured data and facilitate the solving of many graph-related tasks. Bruna et al. [3] first apply the spectral convolution operation to graph-structured data. From the spatial perspective, the spectral convolution can be interpreted to combine the information from its neighbors. GCN [19] simplified this spectral convolution and proposed to use first-order approximation. Since then, many other attention-based, sampling-based and simplified GNN variants which follow the same neighborhood aggregation design have been proposed [37, 14, 12, 41]. Theoretically, those Graph neural networks typically enhance node representations and model expressiveness through a message-passing mechanism, efficiently integrating graph data into the learning of representations [42].

## A.3   Shapley Value in Graph Machine Learning

The Shapley value has found several applications in graph machine learning, primarily in the domain of explainability for Graph Neural Networks. GraphSVX [8] is one of the early works that utilizes the Shapley value to explain the predictions of GNNs. It identifies influential nodes and features for a particular prediction by treating them as players in a cooperative game. However, GraphSVX focuses on local explanations for individual predictions of a fixed GNN. SubgraphX [47] takes a

different approach by explaining GNN predictions through identifying important subgraphs, rather than individual nodes or edges. It uses Monte Carlo tree search to efficiently explore different subgraphs and proposes to use Shapley values as a measure of subgraph importance. EdgeSHAPer [25] is another method that assesses edge importance for GNN predictions using the Shapley value concept. It is particularly relevant for molecular graphs where edges represent chemical bonds. GNNShap [1] extends upon previous Shapley value based GNN explanation methods by providing explanations for edge, leading to better fidelity scores and faster explanations. SAME [44] proposes a structure-aware Shapley-based multipiece explanation method for GNNs that can identify important substructures and provide explanations composed of multiple connected components.

In addition to explainability, Shapley has also been widely adopted for data valuation for conventional machine learning methods as discussed in Section 2. However, it has rarely been utilized for data valuation on graph data. In this work, we pioneer the exploration of graph data valuation, a challenging and previously unexplored problem. Although a recent survey [50] inadvertently refers to GraphSVX as a graph data valuation method, it does not align with the traditional definition of data valuation. We clarify the key differences between graph data valuation (such as our method) and graph explainability (such as GraphSVX) as follows.

1. In general, data valuation (such as our method) aims to understand how graph elements contribute to the model training process, while explainability methods (such as GraphSVX) provide post-hoc explanations for a fixed, pre-trained model.
2. Specifically, our method differs from GraphSVX in several aspects:
   (a) GraphSVX focuses on the explainablity of a local prediction for a single sample, while our method aims to quantify the global contribution of graph elements to the overall model performance.
   (b) GraphSVX operates post hoc, analyzing the contributions of features and nodes in the *testing graph* to the predictions of an already-trained GNN model, while our approach focuses on the global contribution of each data element in the *training graph* to the GNN model's training process.
   (c) GraphSVX employs the standard Shapley value formulation, which assumes free collaboration among players, while our work introduces the `PC-Winter` value to handle the unique hierarchical coalition structures inherent in graph data valuation.

To the best of our knowledge, our investigation constitutes the first foray into graph data valuation, pioneering research in this previously uncharted domain.

# B  Mathematical Formulation of Winter Value

The Shapley value offers a solution for equitable payoff distribution in cooperative games, assuming that players cooperate without any predefined structure. In reality, however, cooperative games often have inherent hierarchical coalitions. To accommodate these structured coalitions, the Winter value extends Shapley value to handle this extra coalition constraints.

Specifically, considering level structures $\mathcal{B}$, with $\mathcal{B} = B_0, \ldots, B_n$ representing a sequence of player partitions. Here, a partition, $B_m$, subdivides the player set $\mathcal{P}$ into a set of disjoint, non-empty subsets $T_1, T_2, \ldots, T_k$. These disjoint subsets satisfy the condition that their union reconstructs the original player set $\mathcal{P}$, which means $T_1 \cup T_2 \cup \ldots \cup T_k = \mathcal{P}$. This partition sequence forms a hierarchy where $B_0$ represents individual players as the leaves of the structure and $B_n$ functions as the root of this hierarchy.

We then determine $\Omega(\mathcal{B})$, the set of all permissible permutations, starting with a single partition $B_m$:

$$\Omega(B_m) = \{\pi \in \Pi(\mathcal{P}) : \forall T \in B_m, \forall i, j \in T \text{ and } k \in \mathcal{P},$$
$$\text{if } \pi(i) < \pi(k) < \pi(j) \text{ then } k \in T\}.$$

$\Omega(\mathcal{B})$ can be further defined as the set of permutations which satisfy all constraints of all levels, $\Omega(\mathcal{B}) = \bigcap_{t=0}^{n} \Omega(B_t)$.

A permissible permutation $\pi$ from the set $\Omega(\mathcal{B})$ requires that players from any derived coalition of $\mathcal{B}$ must appear consecutively. Given the defined set of permissible permutations $\Omega(\mathcal{B})$, the Winter value $\Phi$ for player $i$ is calculated as:

$$\Phi_i(\mathcal{P}, U, \mathcal{B}) = \frac{1}{|\Omega(\mathcal{B})|} \sum_{\pi \in \Omega(\mathcal{B})} \left(U\left(\mathcal{P}_i^\pi \cup i\right) - U\left(\mathcal{P}_i^\pi\right)\right)$$

where $\mathcal{P}_i^\pi = \{j \in N : \pi(j) < \pi(i)\}$ is the set of predecessors of $i$ at the permutation $\sigma$ and $U$ is the utility function in the cooperative game.

## C   Proofs of Theorems

**Theorem 1** (Specificity). *Given a contribution tree $\mathcal{T}$ with a set of players $\mathcal{P}$, any DFS traversal over the $\mathcal{T}$ results in a permissible permutation of $\mathcal{P}$ that satisfies both the Level Constraint and Precedence Constraint.*

*Proof.* We validate the theorem by demonstrating that a permutation obtained through pre-order traversal on $\mathcal{T}$ meets Level Constraints and Precedence Constraints. (1) Level Constraints: During a pre-order traversal of $\mathcal{T}$, a node $p$ and its descendants $\mathcal{D}(p)$ are visited sequentially before moving to another subtree. Thus, in the resulting permutation $\pi$, the positions of $p$ and any $i, j \in \mathcal{D}(p)$ are inherently close to each other, satisfying the condition $|\pi[i] - \pi[j]| \leq |\mathcal{D}(p)|$. This contiguous traversal ensures that all descendants and the node itself form a continuous sequence in $\pi$, meeting the Level Constraint. (2) Precedence Constraints: In the same traversal, each node $p$ is visited before its descendants. Therefore, in $\pi$, the position of $p$ always precedes the positions of its descendants, i.e., $\pi[p] < \pi[i]$ for all $i \in \mathcal{D}(p)$. This traversal pattern naturally embeds the hierarchy of the tree into the permutation, ensuring that ancestors are positioned before their descendants, in line with the Precedence Constraint. $\square$

**Theorem 2** (Exhaustiveness). *Given a contribution tree $\mathcal{T}$ with a set of players $\mathcal{P}$, any permissible permutation $\pi \in \Omega$ can be generated by a corresponding DFS traversal of $\mathcal{T}$.*

*Proof.* To prove the theorem of exhaustiveness, consider a contribution tree $\mathcal{T}$ with a set of players $\mathcal{P}$ and any permissible permutation $\pi \in \Omega$. We apply induction on the depth of $\mathcal{T}$. For the base case, when $\mathcal{T}$ has a depth of 1, which means there are no dependencies among players, any permissible permutation of players is trivially generated by a DFS traversal since there are no constraints on the order of traversal. For the inductive step, assume the theorem holds for contribution trees of depth $k$. For a contribution tree of depth $k+1$ $\mathcal{T}^{k+1}$, consider its root node and subtrees of depth $k$ rooted at the child nodes of the root node. For any given permissible permutation $\pi$ corresponding to the $\mathcal{T}^{k+1}$, according to the Level Constraint, it is a direct composition of the permissible permutations corresponding to the subtrees of depth $k$ rooted at the child nodes of the root node. Now we can construct a DFS traversal over the contribution tree $\mathcal{T}^{k+1}$ that can generate $\pi$. Specifically, the order of composition defines the traversal order of the child nodes of the root node. Furthermore, by the inductive hypothesis, any permissible permutations corresponding to the subtrees can be generated by DFS traversal over the subtrees. Hence, at each child node of the root node, we just follow the corresponding DFS traversal of its corresponding tree. This DFS traversal can generate the given permutation $\pi$, which completes the proof. $\square$

## D   Hierarchical Truncation

In Table 1, we present data comparing the number of model re-trainings on the all six dataset with and without the application of truncation. For the Citeseer dataset, the truncation ratios are defined as 1st-hop: 0.5 and 2nd-hop: 0.7. For the remaining datasets, the truncation ratios are set at 1st-hop: 0.7 and 2nd-hop: 0.9. The results clearly indicate that the number of model re-trainings is substantially reduced when truncation is applied. For instance, focusing on the Citeseer dataset the application of truncation significantly reduces the number of retrainings from 1388 to 535. This significant

decrease, especially in larger datasets like Amazon-Photo and Amazon-Computer, where retraining instances decrease from 147664 to 6258 and from 317959 to 12139 respectively, can be attributed to the substantial number of 2-distance neighbors present in these datasets. The application of truncation effectively reduces the computation by omitting a considerable portion of these neighbors. This finding also implies that overall training time is decreased while still maintaining the ability to accurately measure the total marginal contribution.

Table 1: Retraining Number Comparison Per Permutation

| Dataset | w.o. Truncation | w.t. Truncation |
|---|---|---|
| Cora | 2241 | 756 |
| Citeseer | 1388 | 535 |
| Pubmed | 3683 | 887 |
| Amazon-Photo | 147664 | 6258 |
| Amazon-Computer | 317959 | 12139 |
| Coauther-Physics | 11178 | 852 |

## E   Mixed Node Dropping Experiment

As mentioned in the experiment, labeled nodes will dominate the performance curve when both labeled nodes and unlabeled nodes. The corresponding experiment result is shown in the Figure 6. This experiment validates the assumption that a effective data valuation method would naturally rank labeled nodes for earlier removal over their unlabeled counterparts. For instance, in the Cora dataset, we can observe that the initial drop in accuracy is significant, indicating the removal of high-value labeled nodes. As the experiment progresses and more nodes are removed, the accuracy barely changes, reflecting the removal of unlabeled nodes which has a minimal impact on performance when most labeled nodes are unavailable. The observed pattern across all datasets is consistent: there is a substantial drop in performance at the beginning, followed by a plateau with minimal changes. This suggests that the initial set of nodes removed, predominantly high-value labeled nodes, are those critical to the model's performance, whereas the subsequent nodes show less influence on the outcome.

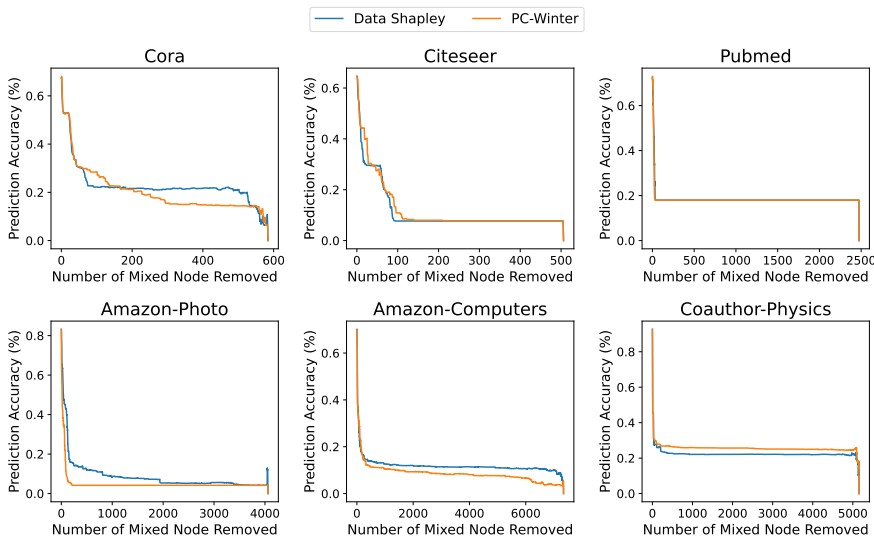

Figure 6: Mixed Node Dropping Experiment

## F  Labeled Node Dropping Experiment

Here, we perform node dropping experiment employing the aggregated value define in the main-body of paper, to demonstrate that `PC-Winter` can capture the heterogeneous influence of labeled nodes. As shown in the Figure 7, both `PC-Winter` and `Data Shapley` demonstrate effectiveness in capturing the diverse contributions of labeled nodes to the model's performance. Particularly in the Pubmed and Amazon-Photo datasets, `PC-Winter` exhibits better performance compared to `Data Shapley`. In other datasets, such as Cora, Citeseer, and Coauthor-Physics, `PC-Winter` shows results that are on par with Data Shapley.

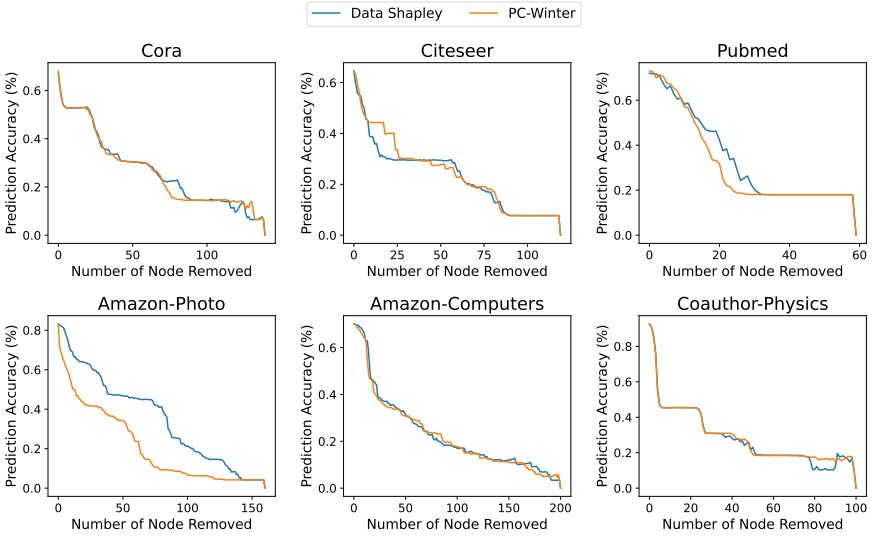

Figure 7: Labeled Node Dropping Experiment

## G  Experimental Details

### G.1  Inductive Setting

Our experiments focus on the inductive node classification task, which aims to generalize a trained model to unseen nodes and is commonly adopted in real-world graph applications [14, 35, 16, 9]. Unlike the transductive setting [19] which incorporates the test nodes in the model training process, the inductive setting separates them apart from the training graph. Such a separation allows us to measure the value of the graph elements in the training graph solely based on their contribution to GNN model training. Following [14], we split each graph $\mathcal{G}$ into 3 disjoint subgraphs: training graph $\mathcal{G}_{tr}$, validation graph $\mathcal{G}_{va}$, and test graph $\mathcal{G}_{te}$. The training graph $\mathcal{G}_{tr}$ is constructed without any nodes from the validation or test set. Correspondingly, edges connecting to a validation node or a testing node are also removed from the training graph. For the validation graph $\mathcal{V}_{va}$ and the testing graph $\mathcal{V}_{te}$, only edges with both nodes within the respective node sets are retained, which is aligned with the inductive setting in prior work [49]. We utilize $\mathcal{G}_{tr}$ to train the GNN model, which is evaluated on $\mathcal{V}_{va}$ for obtaining the data values for elements. The test graph $\mathcal{V}_{te}$ is utilized to evaluate the effectiveness of the obtained values.

### G.2  Datasets

We assess the proposed approach on six real-world benchmark datasets. These include three citation graphs, Cora, Citeseer and Pubmed [30] and two Amazon Datasets, Amazon-Photo and Amazon-Computer, and Coauther-Physics [33]. The detailed statistics of datasets are summarized in Table 2.

Table 2: Dataset Summary

| Dataset | # Node | # Edge | # Class | # Feature | # Train/Val/Test |
|---|---|---|---|---|---|
| Cora | 2,708 | 5,429 | 7 | 1,433 | 140 / 500 / 1,000 |
| Citeseer | 3,327 | 4,732 | 6 | 3,703 | 120 / 500 / 1,000 |
| Pubmed | 19,717 | 44,338 | 3 | 500 | 60 / 500 / 1,000 |
| Amazon-Photo | 7,650 | 119,081 | 8 | 745 | 160 / 20% / 20% |
| Amazon-Computer | 13,752 | 245,861 | 10 | 767 | 200 / 20% / 20% |
| Coauthor-Physics | 34,493 | 247,962 | 8 | 745 | 100 / 20% / 20% |

## G.3 Dataset Split

In the conducted experiments, we split each graph $\mathcal{G}$ into 3 disjoint subgraphs: training graph $\mathcal{G}_{tr}$, validation graph $\mathcal{G}_{va}$, and test graph $\mathcal{G}_{te}$. The training graph $\mathcal{G}_{tr}$ is constructed without any nodes from the validation or test set. Correspondingly, edges connecting to a validation node or a testing node are also removed from the training graph. For the validation graph $\mathcal{V}_{va}$ and the testing graph $\mathcal{V}_{te}$, only edges with both nodes within the respective node sets are retained, which is aligned with the inductive setting in prior work [49]. We utilize $\mathcal{G}_{tr}$ to train the GNN model, which is evaluated on $\mathcal{V}_{va}$ for obtaining the data values for elements. The test graph $\mathcal{V}_{te}$ is utilized to evaluate the effectiveness of the obtained values. In the case of the specific split for each dataset, for the citation networks, we adopt public train/val/test splits in our experiments. For the remaining datasets, we randomly select 20 labeled nodes per class for training, 20% nodes for validation and 20% nodes as the testing set.

## G.4 Convergence Criteria

**Convergence Criterion.** For permutation-based data valuation methods such as `Data Shapley` and `PC-Winter`, we follow convergence criteria similar to the one applied in prior work [13] to determine the number of permutations for approximating data values:

$$\frac{1}{n} \sum_{i=1}^{n} \frac{\left|v_i^t - v_i^{t-20}\right|}{|v_i^t|} < 0.05$$

where $v_i^t$ is the estimated value for the data element $i$ using the first $t$ sampled permutations.

**Time Limit.** For larger datasets, sampling a sufficient number of permutations for converged data values could be impractical in time. To address this and to stay within a realistic scope, we cap the computation time at 120 GPU hours on NVIDIA Titan RTX, after which the calculation is terminated.

## G.5 Truncation Ratios and Hyper-parameters

Table 3 includes the hyper-parameters and truncation ratios used for value estimation.

Table 3: Truncation Ratios and Hyper-parameters

| Dataset | Truncation Ratio | Learning Rate | Epoch | Weight Decay |
|---|---|---|---|---|
| Cora | 0.5-0.7 | 0.01 | 200 | 5e-4 |
| Citeseer | 0.5-0.7 | 0.01 | 200 | 5e-4 |
| Pubmed | 0.5-0.7 | 0.01 | 200 | 5e-4 |
| Amazon-Photo | 0.7-0.9 | 0.1 | 200 | 0 |
| Amazon-Computer | 0.7-0.9 | 0.1 | 200 | 0 |
| Coauthor-Physics | 0.7-0.9 | 0.01 | 30 | 5e-4 |

## G.6 Baselines

### G.6.1 Dropping High-Value Nodes

Here, we introduce the baselines used for comparison to validate the effectiveness of the proposed method in the dropping node experiment:

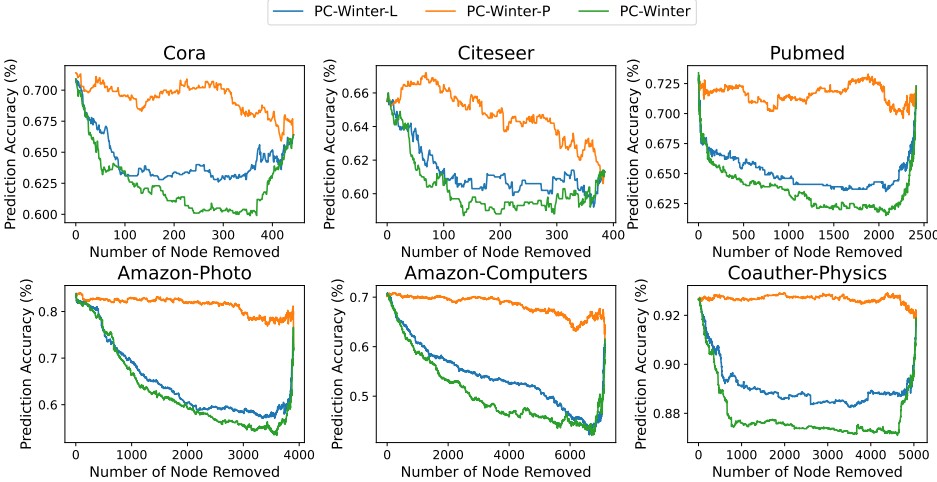

Figure 8: Ablation Study

- `Random Value`: It assigns nodes with random values, which leads to random ranking without any specific pattern or correlation to the node's features.
- `Degree-based Value`: A node is assigned its degree as its value, assuming that a node's importance in the graph is indicated by its degree.
- `Leave-one-out (LOO)`: This method calculates a node's value based on its marginal contribution compared to the rest of the training nodes. Specifically, the value $v(i)$ assigned to each node $i$ is its marginal utility, calculated as $v(i) = U\left(\mathcal{G}_{tr}\right) - U\left(\mathcal{G}_{tr}^{-i}\right)$, where $\mathcal{G}_{tr}^{-i}$ denotes the training graph excluding node $i$. The utility function $U$ measures the model's validation performance when trained on the given graph. In essence, the drop in performance due to the removal of a node is treated as the value of that node.
- `Data Shapley`: The node values are approximated with the Monte Carlo sampling method of `Data Shapley` [13] by treating both labeled nodes and unlabeled nodes as players. Notably, we only include those unlabeled nodes within the 2-hop neighbors of labeled nodes in the evaluation process. There are two approximation methods: Truncated Monte Carlo approximation and Gradient Shapley in [13]. We adopt the Truncated Monte Carlo approximation as it consistently outperforms the other variants in various experiments.

Notably, there is a recent work [6] that aims at characterizing the impact of elements on model performance. Their goal is to approximate `LOO` value. Thus, we do not include it as a baseline as `LOO` is already included.

### G.6.2 Adding High-Value Edges

Here are the detailed descriptions on the baselines applied in the edge adding experiment.

- `Random Value`: it assigns edges with random values, reflecting a baseline where no information are used for differentiating the importance of edges.
- `Edge-Betweeness`: the `Edge-Betweeness` of an edge $e$ is the the fraction of all pairwise shortest paths that go through $e$. This classic approach assesses an edge's importance based on its role in the overall network connectivity.
- `Leave-one-out (LOO)`: This method calculates a edge $e$'s value $v(e)$ based on its marginal contribution compared to the rest of the training graph. In specific, $v(e) = U(\mathcal{G}_{tr}) - U\left(\mathcal{G}_{tr}^{-e}\right)$ Here, $e \in \mathcal{G}_{tr}$ represents an edge in the training graph $\mathcal{G}_{tr}$, and $\mathcal{G}_{tr}^{-e}$ refers to the training graph excluding the edge $e$.

## H Ablation Study and Parameter Analysis

### H.1 Ablation Study

This Appendix Section offers an in-depth ablation analysis across full six datasets to investigates the necessity of both Level Constraint and Precedence Constraint in defining an effective graph value. The results, as shown in Figure 8, consistently demonstrate across all datasets that the absence of either constraint leads to a degraded result when compared to the one incorporating both. This underscores the importance of both two constraints in capturing the contributions of graph elements to overall model performance.

### H.2 The Impact of Permutation Number

This part expands upon the permutation analysis presented in the main paper. It provides comprehensive results across various datasets, illustrating how different numbers of sample permutations impact the accuracy of `PC-Winter`. The results of full datasets are shown in Figure 9. The results reveals

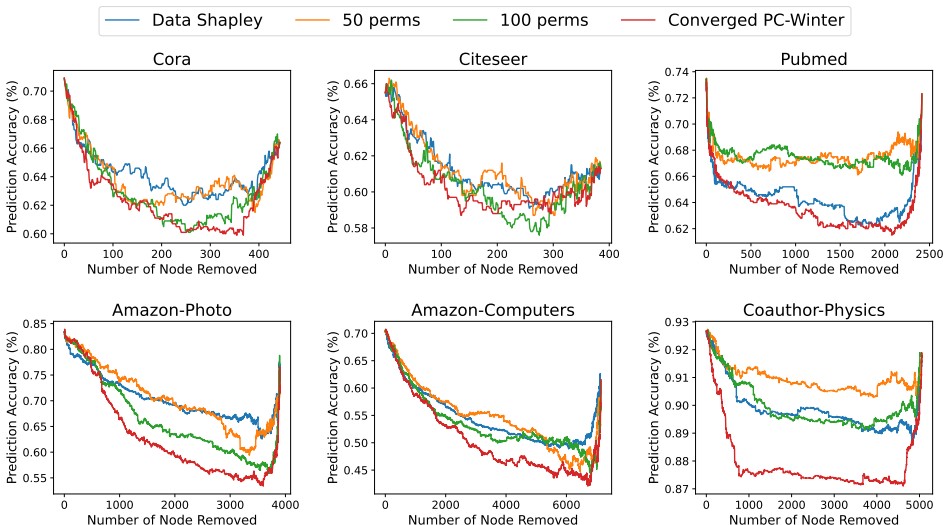

Figure 9: The Impact of Permutation Numbers

that increasing the number of permutations generally improves the performance and accuracy of the valuation. `PC-Winter` also show robust results even with a limited number of permutations, highlighting its effectiveness. The phenomenon is consistent across all datasets where our approach with just 50 to 100 permutations manages to compete closely with the fully converged `Data Shapley`, emphasizing the efficiency of `PC-Winter` in various settings.

### H.3 The Impact of Truncation Ratios

Our approach involves truncating the iterations involving the first and second-hop neighbors of a labeled node during value estimation. Here, we investigate the impact of truncation proportion on overall performance, using the same number of permutations as in our primary node-dropping experiment. As shown in Figure 10, we adjusted the truncation ratios for the Citation Network datasets. The ratios ranged from truncating 50% of the first-hop and 70% of the second-hop neighbors (0.5-0.7), up to 90% truncation for either first-hop (0.9-0.7) or second-hop (0.5-0.9) neighbors. For the Cora and Citeseer datasets, increasing truncation at the first-hop level had a minimal impact on performance, and `PC-Winter` still significantly outperformed `Data Shapley`. In the case of the Pubmed dataset, more extensive truncation at the first-hop level notably reduced performance. Regarding large datasets such as the Amazon, while truncation at either the first or second-hop levels had a marginal negative effect on performance, `PC-Winter` 's estimated data values generally remained superior to results of `Data Shapley`.

In addition, we provide a detailed analysis of our truncation strategy across other datasets. It includes results not presented in the main text, focusing on the impact of limiting model retraining times to the first and second-hop neighbors in value estimation. We investigate the impact of truncation proportion on overall performance, using the same number of permutations as in our primary node-dropping experiment. The findings on full datasets are illustrated in Figure 10. Specifically, our findings reveal that in datasets like Cora and Citeseer, adjusting truncation primarily at the first-hop level has a negligible impact on the accuracy of node valuation, with `PC-Winter` still maintaining a considerable advantage over `Data Shapley`. For large datasets such as the Amazon-Photo, Amazon-Computers and Coauther-Physics, while truncations had a marginal negative effect on performance, `PC-Winter`'s estimated data values generally remained better than Data Shapley. This analysis indicates that `PC-Winter` can afford to employ larger truncation, enhancing computational efficiency without substantially sacrificing the quality of data valuation.

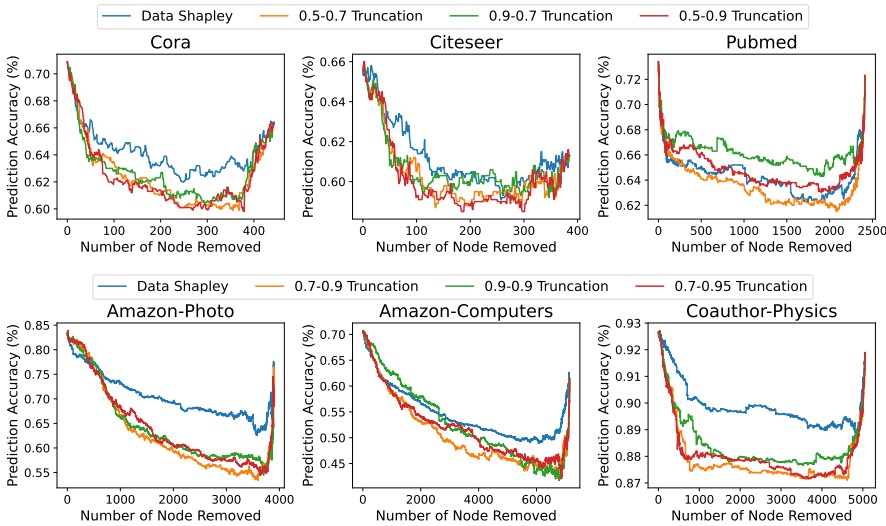

Figure 10: The Impact of Truncation Ratios

Table 4: Permutation Number and Time Comparison

| Dataset | Truncation | PC-Winter | | Data Shapley | |
|---|---|---|---|---|---|
| | | **Perm Number** | **Perm Time (hrs)** | **Perm Number** | **Perm Time (hrs)** |
| Cora | 0.5-0.7 | 325 | 0.013 | 327 | 0.024 |
| Citeseer | 0.5-0.7 | 291 | 0.018 | 279 | 0.037 |
| Pubmed | 0.5-0.7 | 316 | 0.025 | 281 | 0.285 |
| Amazon-Photo | 0.7-0.9 | 418 | 0.211 | 109 | 1.105 |
| Amazon-Computer | 0.7-0.9 | 181 | 0.662 | 33 | 3.566 |
| Coauthor-Physics | 0.7-0.9 | 460 | 0.119 | 45 | 2.642 |

## H.4 Efficiency Analysis

Here, we compare the computational efficiency of our proposed method `PC-Winter` and the `Data Shapley` approach in terms of permutation number and time per permutation. As detailed in Table 4, the results indicate that `PC-Winter` requires significantly less time to compute each permutation across various datasets. Specifically, for the Cora dataset, `PC-Winter` completes each permutation in approximately half the time required by `Data Shapley`. Moving to larger datasets, the efficiency of `PC-Winter` becomes even more pronounced. For instance, in the Amazon-Computer dataset, `PC-Winter`'s permutation time is only a fraction of what is required by `Data Shapley` —`PC-Winter` takes slightly over half an hour per permutation whereas `Data Shapley` exceeds three

and a half hours. This consistent reduction in permutation time demonstrates the computational advantage of `PC-Winter`, particularly when handling large graphs. Combining the insights from the Permutation Analysis shown in Figure 9 with the Permutation Comparison Table 4, we observe that for datasets such as Cora, Citeseer, Amazon-Photo, and Amazon-Computer, around 50 permutations are sufficient for `PC-Winter` to achieve performance comparable to that of `Data Shapley`. Simple calculations demonstrate that our method is significantly faster than `Data Shapley` in achieving similar performance levels. For instance, in the Cora dataset, the speedup factor is $\frac{327 \times 0.024}{50 \times 0.013} = 12.07$, and for the Citeseer dataset, it is $\frac{279 \times 0.037}{50 \times 0.018} = 11.47$. The speedup factors for Amazon-Photo and Amazon-Computer are $\frac{109 \times 1.105}{50 \times 0.211} = 11.42$, and $\frac{33 \times 3.566}{50 \times 0.662} = 3.57$, respectively. For Coauthor-Physics, it takes about 100 permutations for `PC-Winter` to match the performance of `Data Shapley`, which implies a speedup factor of $\frac{45 \times 2.642}{100 \times 0.119} = 10.00$. In conclusion, `PC-Winter` can achieve stronger performance than `Data Shapley` using the same or even less time. Furthermore, it takes `PC-Winter` much less time to achieve comparable performance as `Data Shapley`. Notably, though `PC-Winter` is significantly more efficient than `Data Shapley`, its scalability is still limited, and future work in further improving its efficiency is desired.

## H.5 Complexity Analysis

We analyze the complexity of the PC-Winter. For convenience, we assume that we are dealing with a $d$-regular graph. There are a total of $L$ labeled nodes in the graph. As described in the paper, we deal with a GNN model with 2 layers. Without loss of generality, we use $F$ to denote the dimensionality of node representations in each layer. We assume the number of classes in the dataset is $C$. For hierarchical truncation, we assume we adopt a truncation ratio of $r_1 - r_2$, which is consistent with the description in Section 3.4. Then, the number of nodes in a computation tree for any labeled node is $N_{\text{full}} = 1 + d + d^2$. With hierarchical truncation, the number of nodes in the truncated computation tree is $N_{trun} = 1 + d \cdot (1 - r_1) + d^2 \cdot (1 - r_1)(1 - r_2)$. When the truncation ratios are large, $N_{trun} \ll N_{full}$. For instance, when $r_1 = r_2 = 0.9$, $N_{trun}$ could be less than 5Time Complexity Analysis: We now analyze the time complexity of a single permissible permutation of the PC-Winter algorithm. We begin by examining the time complexity of generating a single permissive permutation. Then, we investigate the complexity of a single model retraining and provide the total retraining number for a single permutation. Finally, we combine these analyses to derive the overall time complexity for generating one permissible permutation and going through it for calculating the marginal contributions. Time complexity of generating a single permissive permutation: The time complexity of traversing the truncated contribution tree to generate a single permissive permutation is $O(L \cdot N_{\text{trun}})$. In particular, there are $L \cdot N_{trun} + 1$ nodes in the contribution tree (including the dummy node). Hence, the cost of a DFS traversal over the contribution tree is $O(L \cdot N_{\text{trun}} + 1 + L \cdot N_{\text{trun}})$ $= O(L \cdot N_{\text{trun}})$. Time complexity of one model retraining: As described in Section 3.4, with local propagation, for each model retraining, we only need to perform feature aggregation on a single partial computation tree. The size of a partial computation tree is, on average, $\frac{N_{\text{trun}}}{2}$. Therefore, the feature aggregation complexity for each retraining step is $O(\frac{N_{\text{trun}}}{2} \cdot F)$, where $F$ is the dimension of node features. The feature transformation complexity for each model retraining is $O(F \cdot F + F \cdot C) = O(F^2)$, where $C$ is the output dimension (number of classes) of the GNN model. Therefore, the total time complexity of a single retraining is $O(\frac{N_{\text{trun}}}{2} \cdot F + F^2)$. Without local propagation, the feature aggregation complexity for each model retraining would be much larger, since the propagation needs to be performed on the entire graph. The number of model retraining in a single permutation: In a permissible permutation, we need to perform retraining for each node in the truncated contribution tree, which has $L \cdot N_{\text{trun}}$ nodes in total. Therefore, $L \cdot N_{\text{trun}}$ model retrainings are needed for a single permutation. Total time complexity for a single permissible permutation: With local propagation and hierarchical truncation, the total time complexity of a single permissible permutation in PC-Winter is: $O(L \cdot N_{\text{trun}} + L \cdot N_{\text{trun}} \cdot (\frac{N_{\text{trun}}}{2} \cdot F + F^2)) = O(L \cdot N_{\text{trun}} \cdot (1 + \frac{N_{\text{trun}}}{2} \cdot F + F^2)) = O(L \cdot N_{\text{trun}} \cdot (\frac{N_{\text{trun}}}{2} \cdot F + F^2))$. Notably, the time complexity of generating a permissible permutation is negligible compared to the cost of model retraining. The proposed strategies, hierarchical truncation, and local propagation, help reduce the overall time complexity of the PC-Winter algorithm. In particular, hierarchical truncation makes $N_{\text{trun}}$ much smaller than $N_{\text{full}}$, greatly decreasing the total number of model retraining required

for a single permutation. On the other hand, Local propagation reduces the feature aggregation complexity, greatly reducing the cost of each retraining.

## H.6 Code Availability

To facilitate the reproducibility of our work and to encourage further research in the field of graph data valuation, we have made our code publicly available on an anonymous repository at `https://anonymous.4open.science/r/graph-data-valuation-B348`. The repository contains the implementation of the `PC-Winter` algorithm, along with scripts for running the experiments presented in this paper. We welcome researchers and practitioners to utilize and build upon our code for their own research and applications in graph data valuation.

