# OpenReview forum: "Precedence-Constrained Winter Value for Effective Graph Data Valuation"
_NeurIPS.cc/2024/Datasets_and_Benchmarks_Track — Submitted to NeurIPS 2024 Track Datasets and Benchmarks_

### Official Review · Reviewer_DXbo · 2024-07-08
**Summary This paper discusses how nodes in a hierarchical graph contribute to model performance and the challenges of graph data valuation**

**Rating:** 5
**Confidence:** 3
**Correctness:** yep
**Clarity:** yep

**Review:**

Strengths
1.	The paper is written clearly and all required background is well-explained.
2.	The problem of graph data valuation is important and impactful. Facing the special attributes of graph-structured data, the author delved into how labeled and unlabeled nodes contribute to the results, formalizing this problem using special coalition structures.
3.	The paper proposes a clever approach to using DFS to obtain permutations, as well as efficiently estimate PC-Winter through hierarchical truncation and local aggregation. These methods make graph data valuation feasible with neural networks.
4.	The paper provides detailed analyses from multiple perspectives.

Weaknesses
1.	It would be best to explain why there are only two truncation ratios for 2-distance neighbors.
2.	As a core contribution, efficiency evaluation should receive more attention.
3.	Figure 1 might be unnecessary.
4.	Does the effectiveness of the proposed efficient approximation method get affected by the quantity of nodes and edges? It is better to discuss this.

**Strengths:**

Pls see Review

**Additional Feedback:**

na

**Documentation:**

yep

**Limitations:**

yep

**Opportunities For Improvement:**

1.	There are only two truncation ratios in the experiment and all examples in this paper have only three levels. Does this mean the height of the computational tree will be limited? If the influence of >2-distance neighbors is minimal or these nodes are omitted, perhaps it should be stated.
2.	The efficient estimation of the PC-Winter proposed in the paper is an important contribution. However, there are few relevant evaluations in the main body (mostly in the Appendix), so having at least one table would be good. Also, efficient estimation of Shapley value has been widely studied (e.g., [1][2]), can they be adapted to compute PC-Winter and be used as baselines to demonstrate the superiority of the proposed method?
[1] Ruoxi Jia, David Dao, Boxin Wang, Frances Ann Hubis, Nick Hynes, Nezihe Merve Gürel, Bo Li, Ce Zhang, Dawn Song, and Costas J. Spanos. 2019. Towards Efficient Data Valuation Based on the Shapley Value.
[2] Jiayao Zhang, Qiheng Sun, Jinfei Liu, Li Xiong, Jian Pei, and Kui Ren. 2023. Efficient Sampling Approaches to Shapley Value Approximation.
3.	Figure 1 resulted in a strange format and looked similar to Figure 2(b). Is it possible to describe the winter value based on Figure 2(b)?
4.	Is there a correlation between the number of nodes and edges and the improvement in efficiency? We noticed that the number of nodes in the Amazon-Photo dataset is about 6% of the number of edges, with an efficiency improvement of about 26%. Meanwhile, in the Coauthor-Physics dataset, the number of nodes is about 14% of the number of edges, resulting in an efficiency improvement of about 50%

**Relation To Prior Work:**

yep

**Summary And Contributions:**

This paper discusses how nodes in a hierarchical graph contribute to model performance and the challenges of graph data valuation. Based on the Winter value, it proposes PC-Winter, which constrains the sampling permutations due to the Level Coalition Structure and the Unilateral Dependency Structure. Then, this paper proposes hierarchical truncation and local aggregation for efficient approximation of PC-Winter. The paper provides extensive experiments and thorough analysis.
I wonder if this paper suits the Track of Datasets and Benchmark.

---

> ### Author Rebuttal · Authors · 2024-08-19
>
> **Q1. "It would be best to explain why there are only two truncation ratios for 2-distance neighbors. There are only two truncation ratios in the experiment and all examples in this paper have only three levels. Does this mean the height of the computational tree will be limited? If the influence of >2-distance neighbors is minimal or these nodes are omitted, perhaps it should be stated."**
>
> A1: We appreciate the reviewer's insightful question about our truncation ratios. Our choice of two truncation ratios corresponds to the use of 2-layer GNN models in our experiments. Notably, 2-layer GNNs are commonly adopted in the literature and generally provide great overall performance for many tasks [1-3]. In a 2-layer GNN, the computation tree for each node effectively spans two hops in the graph. This means that for any given node, only its 1-hop and 2-hop neighbors contribute to its final representation and, consequently, to the model's performance. Any neighbors beyond 2 hops will have no contribution and therefore will be assigned a value of $0$ even if we take them into consideration. Thus, with 2-layer GNNs, the corresponding computational tree has a limited height, requiring only two truncation ratios--one for each layer. We will clearly state and further clarify these points in the revision. However, it's important to note that our method is flexible and can be extended to accommodate GNNs with more layers. In such cases, we can introduce additional truncation ratio coefficients for each layer. In this scenario, it is still possible that the influence of '>2-distance' neighbors is minimal. If so, we may still disregard '>2-distance' neighbors during the valuation and only consider '<2-distance' neighbors with 2 truncation ratios. We leave such investigations as future work.
>
> [1] Kipf, Thomas N., and Max Welling. "Semi-supervised classification with graph convolutional networks."
> [2] Wu, Felix, et al. "Simplifying graph convolutional networks."
> [3] Veličković, Petar, et al. "Graph attention networks."
>
> ---
>
> **Q2."As a core contribution, efficiency evaluation should receive more attention. The efficient estimation of the PC-Winter proposed in the paper is an important contribution. However, there are few relevant evaluations in the main body (mostly in the Appendix), so having at least one table would be good."**
>
> A2: Thank you for your suggestion. We agree that efficiency evaluation is a core component of our work. We will move the relevant results from the Appendix into the main body of the paper and include a table to highlight the efficiency evaluation more clearly.
>
> ---
>
> **Q3."Also, efficient estimation of Shapley value has been widely studied (e.g., [1][2]), can they be adapted to compute PC-Winter and be used as baselines to demonstrate the superiority of the proposed method?"**
>
> A3:
> We appreciate the reviewer's suggestion to consider existing Shapley value estimation methods. We acknowledge some of them could potentially be adapted to improve PC-Winter's efficiency. However, we would like to clarify that these strategies are complementary to our approach, as they improve efficiency from different perspectives. Therefore, we are not competing with these methods; rather, we could potentially combine them with our approach. We leave the exploration of such combinations as future work. Next, we describe and compare these different strategies and then discuss how the existing strategies complement our proposed strategies.
>
> - Our strategies (PC-Winter itself, Hierarchical Truncation, and Local Propagation) are specifically designed to leverage graph information, effectively reducing the definitive space of permutations, decreasing the needed computations in the graph context, and accelerating feature propagation in GNNs. These are tailored to address the unique challenges posed by graph-structured data. Notably, our method does not include optimization related to speeding up model retraining.
>
> - On the other hand, existing strategies, such as those mentioned by the reviewer, focus on other more general aspects of efficiency improvement: (1) Influence function-based approaches [1]: These methods approximate parameter changes after adding or removing data points, reducing the time complexity for model retraining. They use influence functions to estimate the impact of individual data points on model parameters efficiently. (2) Utility learning methods [2]: These approaches learn a surrogate model of the utility function, allowing for quick estimation of utilities for large numbers of data subsets without repeated model training. In summary, both of these strategies aim to estimate utility in an efficient way.
>
> These existing strategies to improve the efficiency of Shapley are complementary to our approaches. Specifically, even with our strategies, we can still utilize influence functions or utility learning for more efficient utility estimation. However, it is important to note that the efficiency gains of the aforementioned influence function and utility learning often come at the cost of accuracy in utility estimation, which may negatively impact the value estimation. Therefore, in this paper, to better evaluate the effectiveness brought by the algorithmic design of PC-Winter, we do not adopt these strategies. However, we believe that exploring their integration with PC-Winter could lead to even more efficient graph data valuation methods, and we plan to investigate this potential improvement in future work.
>
>
> [1] Ruoxi Jia, David Dao, Boxin Wang, Frances Ann Hubis, Nick Hynes, Nezihe Merve Gürel, Bo Li, Ce Zhang, Dawn Song, and Costas J. Spanos. "Towards Efficient Data Valuation Based on the Shapley Value."
>
> [2] Wang, Tianhao, Yu Yang, and Ruoxi Jia. "Improving cooperative game theory-based data valuation via data utility learning."
>
> ---

---

> > ### Author Rebuttal · Authors · 2024-08-19
> >
> > **Q4."Figure 1 resulted in a strange format and looked similar to Figure 2(b). Is it possible to describe the winter value based on Figure 2(b)?"**
> >
> > A4: We appreciate the reviewer's observation regarding the similarity between Figures 1 and 2(b). Figure 1 illustrates the Level Coalition Structure as discussed in Section 2.1. On the other hand, Figure 2(b) visually represents the coalition structure of the graph data valuation game, which incorporates both the Level Coalition Structure and the Unilateral Dependency Structure. In particular, in Figure 2(b), the nodes highlighted in red are the "delegates" in each sub-coalition. The "delegate" is an essential concept in the Unilateral Dependency Structure (see Lines 187-197 and Observation 3). Therefore, if we disregard the red highlights in Figure 2(b), we could indeed use Figure 2(b) to illustrate the Winter value.
> >
> > We will revise these figures and optimize the description texts in our next iteration to make the illustration clearer. Thank you for helping us enhance the presentation of our work.
> >
> > ---
> >
> > **Q5. "Is there a correlation between the number of nodes and edges and the improvement in efficiency? We noticed that the number of nodes in the Amazon-Photo dataset is about 6% of the number of edges, with an efficiency improvement of about 26%. Meanwhile, in the Coauthor-Physics dataset, the number of nodes is about 14% of the number of edges, resulting in an efficiency improvement of about 50%"**
> >
> > A5: We appreciate the reviewer's observation regarding the relationship between graph structure and efficiency improvements. Indeed, we have observed a correlation between graph structure and efficiency gains. In particular, we have conducted a detailed time complexity analysis in Appendix H.5. For convenience, the analysis is conducted for a d-regular graph, where each node has a degree $d$. In the analysis, the time complexity of PC-Winter for a single permutation is $O(L \cdot N_{trun} \cdot (N_{trun}/2 \cdot F + F^2))$, where $N_{trun} = 1 + d \cdot (1 - r_1) + d^2 \cdot (1 - r_1)(1 - r_2)$. Here, $L$ is the number of labeled nodes, $F$ is the dimensionality of node features in each layer of the GNN model, and $r_1$ and $r_2$ are the truncation ratios. The degree $d$ directly reflects the ratio between nodes and edges. Specifically, a smaller $d$ indicates a larger node-to-edge ratio (#nodes/#edges). According to this analysis, graphs with a higher ratio of nodes to edges (closer to tree structures, i.e., with smaller $d$) have higher efficiency gains. Though the time complexity analysis is for d-regular graphs, the conclusions generally hold true for all graphs. This analysis explains the observed correlation between graph structure and efficiency improvements, with sparser graphs (higher node-to-edge ratio) generally showing greater efficiency gains from our method.
> >
> > ---

---

> > > ### Author Response · Authors · 2024-08-26
> > > **A kind reminder**
> > >
> > > Thank you for taking the time to review our work. We appreciate your feedback and we have prepared a thorough response to address your concerns. We believe that we have responded to and addressed all your concerns — in light of this, we hope you consider raising your score. Feel free to let us know in case there are outstanding concerns, and if so, we will be happy to respond.
> > >
> > > Notably, given that we are approaching the deadline for the rebuttal phase, we hope we can have the discussion soon. Thanks!

---

> ### Comment · Area_Chair_bK4C · 2024-08-29
> **Please respond/acknowledge author rebuttals**
>
> Dear Reviewer,
>
> As the AC, I'd like to remind you to respond/acknowledge the authors' rebuttal of this paper. Only three days left till the end of the discussion period!
>
> Thanks for your help and participation!

---

### Official Review · Reviewer_2sH2 · 2024-07-21
**Strong contribution to data valuation literature that tackles an unexplored area (graph data). Provides theoretical basis and strong experiments.**

**Rating:** 9
**Confidence:** 4

**Review:**

Overall, this is a very strong paper that makes a large contribution to the subfield of data valuation. While potentially dense depending on the reader's familiarity (with Shapley value for ML literature, with graph neural networks), it is well-organized and clear throughout. The experiments are very compelling and the appendix answers many potential robustness-related questions.

The main caveat one might have with this paper is that it is not directly a "Dataset" or "Benchmark" contribution, though I would argue that "core data valuation work" belongs in this track (the results do serve as a kind of benchmark for valuation tasks).

**Strengths:**

### Significance
The paper claims to add a major new perspective to data valuation discussions, and I think the proposed valuation approach and accompanying experiments back up the strong claims of the paper (that this represents a first stab at an unexplored area, and that this new method will provide real value for practitioners using GNNs).

### Relevance to broad research community
This paper will be of immediate interest primarily to the data valuation community and to researchers working with GNNs. Overall, given the use of such techniques across many domains and the fact that data valuation may become more widely used in general, I expect this to have some broad impact in the long run. Overall: targeted impact in the near term, potentially broad impact.
### Quality
Clarity is very good overall given the contribution (see a few additional comments below).

The experiments are well-designed and presented. I had some questions as a reader upon reading the main body and found they were answered in the Appendix. I think the current organization of content between main and appendix is reasonable, although some re-organization could make sense (in particular -- more details about dataset in main body).
### Ethical and social implications
The social implications of this work (using value estimates for remuneration and credit) are briefly alluded to, but not much discussed. I think the level of engagement is reasonable given that this core data valuation work.

**Additional Feedback:**

A few very minor comments that the authors may already be aware of:
- Some figures are oddly placed and thus not as helpful to reader as they could be. I would emphasize them a bit more in next iteration.
- Recommend adding some paragraph breaks to Appendix H.5

**Clarity:**

Clarity of writing is good throughout. Most of the suggestions above highlight some opportunities to add even more clarity, mainly around situating graph data and GNNs relative to prior work on data valuation for non-graphical / non-linked data.

I expect readers familiar with Shapley values and GNNs will find this paper very straightforward. Those familiar with one or the other may find this to be a kind of paper that might require a few reads and/or jumping between main and appendix, but this is reasonable overall and nothing obvious jumps out in terms of major reorganization or any specific claims or definitions that are unclear.

**Correctness:**

The experiments are well executed and are likely to account for many questions potential users of the this technique might have.

**Documentation:**

As noted above, not a dataset or benchmark per se. Description of the method is good. Experiments should be reproducible.

**Ethics:**

No major concerns. With more space, the draft could discuss social implications, especially around payment, data economics, etc. in more detail.

**Limitations:**

Minor suggestion: the paper could stand to benefit slightly from an explicit limitations section that focuses on *why practitioners might not use data values in practice*.

**Opportunities For Improvement:**

One area for potential improvement in the next iteration of this paper is to further increase the clarity along several fronts. Some of these are not about describing the method per se:

#### Further clarity about how people might use PC-Winter and "traditional" data value estimates in practice

- Clarify (even just intuitively) the interaction between PC-Winter Values and traditional "Euclidean" data values such as Shapley for data that could be treated as graph or non-graph. For instance, as I understand here, all of the datasets here could be treated as non-graphical data (just omit the identifiers used to create nodes and edges) and a number of classification tasks could be explored.
- Very practically, based on these experiments will practitioners still want to calculate both? It seems one simple interpretation is that these results are mainly useful if one is already using GNNs (presumably, this means using information from neighbours of an item provides a performance advantage). In this case, should we expect PC-Winter to make data values obtained on non-graph tasks relatively "obsolete"?
- Note: these questions may be out of scope (more along the lines of future work), but may help to touch on briefly. These questions also relate to a broader question about when to use GNNs, how to enrich non-graphical data with edge information, etc.

#### Provide example of archetypical task
In the current draft, the datasets are very briefly explained (with full details in the Appendix). While reasonable for readers familiar with GNN literature, I think using some space in the main body to walk through an example task and example observation -- with a brief natural language of how the PC-Winter Value might be interpreted in the context of this task -- could go a very long way in increasing impact outside of those using GNNs (though perhaps this work will help to expand the pool of GNN users in the long run!)

This is, overall, a paper that could benefit from a running example. In terms of increasing "speed-to-grok" of readers, it would be especially helpful if the running example mapped to the "org chart" example used to explain hierarchical coalitions and the idea of a contribution tree and "permissible permutations" of "players".

**Relation To Prior Work:**

The discussion of prior work was very good. I think the current draft makes a reasonable choice to walk through prior work on Shapley values and GNNs. This will overall increase the reader pool and impact. Appendix further helped out.

**Summary And Contributions:**

This paper presents a new data value estimation approach: precedence constrained winter values. The draft motivates and defines this approach, then presents a variety of experiments showing that value estimates produced using PC-Winter are particularly useful for graph neural networks that leverage graph-structured data (contrasting this with existing methods that target "Euclidean" data for supervised learning).

---

> ### Author Rebuttal · Authors · 2024-08-19
>
> **Q1. "Clarify (even just intuitively) the interaction between PC-Winter Values and traditional "Euclidean" data values such as Shapley for data that could be treated as graph or non-graph. For instance, as I understand here, all of the datasets here could be treated as non-graphical data (just omit the identifiers used to create nodes and edges) and a number of classification tasks could be explored."**
>
> A1.
>  Thank you for your insightful question on the relationship between PC-Winter and traditional data valuation methods. We appreciate the opportunity to clarify this point. We understand the question as asking how PC-Winter and Data Shapley compare when graph data is transformed into non-graph data by omitting the identifiers used to create nodes and edges. If we have misunderstood, please feel free to correct us.
>
> Indeed, as the reviewer points out, all of the datasets we use could be treated as non-graphical data by omitting the identifiers used to create nodes and edges. In this scenario, where we treat graph data as "non-graph data" for node valuation, all nodes would effectively become isolated data points.
> When graph data is treated as non-graph data, PC-Winter naturally becomes Data Shapley as detailed below.
>
> * Without graph structures, there are no Level Coalition Structure or Unilateral Dependency Structure as defined in our paper (Section 3.2).
> * Consequently, the two constraints we introduced - the Level Constraint and the Precedence Constraint (Constraints 1 and 2 in Section 3.2) - are no longer applicable.
> * As a result, the set of permissible permutations $Ω$ in PC-Winter (Equation 3) becomes equivalent to the set of all possible permutations $Π$ used in traditional Shapley values (Equation 1).
> * The utility function would simplify to evaluating model performance on individual data points, similar to traditional Shapley value calculations.
>
> Therefore, in this scenario, PC-Winter's valuation process would mirror that of traditional Shapley values, calculating the average marginal contribution of each data point across all permutations.
>
> ---
> **Q2.  "Very practically, based on these experiments will practitioners still want to calculate both? It seems one simple interpretation is that these results are mainly useful if one is already using GNNs (presumably, this means using information from neighbours of an item provides a performance advantage). In this case, should we expect PC-Winter to make data values obtained on non-graph tasks relatively "obsolete"?"**
>
> A2. We appreciate the chance to discuss the applicability of PC-Winter in different scenarios. We interpret your question as asking in which scenarios PC-Winter is more suitable, and what impact PC-Winter might have when used for valuation in non-graph-based models. If we have misunderstood, please feel free to correct us.
>
> We agree that PC-Winter is particularly advantageous in scenarios with useful graph information or when using GNN models. In these cases, PC-Winter can indeed make data values obtained on graph tasks more accurate and informative compared to traditional methods.
>
> However, if we consider whether PC-Winter would make data values obtained on non-graph tasks "obsolete", the situation is less clear-cut. For instance, if we have graph information but are performing a non-graph task (e.g., only using isolated node features to train an MLP for prediction and we want to know node values), it's difficult to determine whether the graph-informed permutations of PC-Winter or the completely random permutations of traditional Shapley would produce better data value estimates. Therefore, if we care both graph-related tasks and non-graph tasks on a given graph data, it might be helpful for us to obtain both data values. We acknowledge that this is an area worth further exploration and plan to investigate it in future work.
>
> ---
> **Q3. "In the current draft, the datasets are very briefly explained (with full details in the Appendix). While reasonable for readers familiar with GNN literature, I think using some space in the main body to walk through an example task and example observation -- with a brief natural language of how the PC-Winter Value might be interpreted in the context of this task -- could go a very long way in increasing impact outside of those using GNNs"**
>
> A3. Thank you for the suggestion to provide a more detailed and concrete example! We agree that this would enhance the accessibility of our work to a broader audience. We will attempt to design such an example for better illustration.
>
> ---
>
> **Q4."Minor suggestion: the paper could stand to benefit slightly from an explicit limitations section that focuses on why practitioners might not use data values in practice."**
>
> A4: Thank you for the suggestion to add an explicit limitations section focusing on practical challenges in using data values. We agree this will enhance the paper's balance and applicability. Here are some key limitations and challenges practitioners may face when applying PC-Winter or data valuation methods in general:
>
> 1. Computational Complexity: Despite our efficiency improvements, calculating PC-Winter values for very large graphs remains computationally intensive. This may limit its applicability in scenarios with strict time constraints or for extremely large-scale graphs.
>
> 2. Limited Applicability to Heterogeneous Graphs: PC-Winter was originally designed with homogeneous graphs in mind. When applied to heterogeneous graphs, some information about node and edge types would need to be omitted or simplified, which could potentially impact performance. The method might not fully capture the complex interactions between different node and edge types, leading to less accurate valuations. Further work is needed to validate and potentially extend PC-Winter's performance on heterogeneous graph structures.
>
> ---

---

> > ### Author Rebuttal · Authors · 2024-08-19
> >
> > **Q5."Some figures are oddly placed and thus not as helpful to reader as they could be. I would emphasize them a bit more in next iteration."**
> >
> > A5: Thank you for pointing this out. We will improve the placement and emphasis of our figures in the next iteration to enhance readability and clarity.
> >
> > ---
> >
> > **Q6. "Recommend adding some paragraph breaks to Appendix H.5"**
> >
> > A6: We appreciate your attention to detail. We will add paragraph breaks to Appendix H.5 to improve its readability and structure.
> >
> > ---

---

> > > ### Author Response · Authors · 2024-08-26
> > > **Thank you for your review**
> > >
> > > Thank you for your thorough review and positive feedback on our work. We appreciate your recognition of the significance and relevance of our contribution to the field of data valuation, particularly in the context of graph-structured data. We believe we have addressed these points in our detailed rebuttal. However, if you feel any aspect requires further clarification or elaboration, please don't hesitate to let us know.

---

### Official Review · Reviewer_deQC · 2024-07-25
**An interesting technique with limited experimental validation.**

**Rating:** 7
**Confidence:** 4

**Review:**

### Summary
The manuscript proposes a theoretically well-founded method to tackle an interesting problem in data-centric graph learning.
While the suggested approach appears promising, its experimental evaluation is limited, and the discussion of limitations and potential adverse impacts is lacking.
Unless these weaknesses are addressed, I cannot support publication on the NeurIPS Datasets and Benchmarks track.

### Strengths
- (S1) Important and relevant problem (graph-data valuation)
- (S2) Sound game-theoretic foundation of proposed data-valuation measure (PC-Winter)

### Weaknesses
- (W1) Limited experimental evaluation (see "Correctness")
- (W2) No/insufficient discussion of potential negative impacts and limitations (see "Limitations")

**Strengths:**

See above

**Additional Feedback:**

- Observation 1 (ll. 133–140) seems redundant, given the challenges you state in the introduction.
- The first line of the abstract is tautological: "quantifying data's worth" (better: "the worth of data") is just another way of saying "data valuation".
- Please check for typos (e.g., l. 319: "Coauther" -> "Coauthor") and missing hyphens (e.g., "graph data valuation" -> "graph-data valuation", "data valuation frameworks" -> "data-valuation frameworks"), and please use "of" to express possessive forms for non-human entities (e.g., "data's" -> "of the data", "graph's" -> "of the graph").
- Please increase the spacing around wrap-around figures.
- Please ensure that all symbols used in the main text are also defined in the main text (e.g., G_{tr}, G_{te}).
- l. 222 and Eq. 3 (probably also in other places): There seem to be curly braces missing around p in "D(p) \cup \{p\}".

**Clarity:**

The paper is mostly easy to follow, and I ppreciate the frequent main-text pointers to the relevant appendix sections.
However, the Experiments section is confusing (even when read in conjunction with the more elaborate description in the appendix).
For example:

> We utilize G_{tr} to train the GNN model, which is evaluated on V_{va} (G_{va}?) for obtaining the data values for elements. The test graph V_{te} (G_{te}?) is utilized to evaluate the effectiveness of the obtained values.

Can you clarify what this means?
A small drawing depicting your experimental process on a toy graph might be particularly helpful.

**Correctness:**

While the authors conduct many experiments, these experiments do not sufficiently support their claims.

In particular:
- The evaluation is conducted using only one GNN variant (please also provide a high-level description of the architecture in the main text and a detailed specification in the appendix).
- The evaluation tasks are limited to experiments on inductive node classification (despite the abstract promising "diverse datasets and tasks").
- It is unclear how robust the data valuations are to noise in the data (this could be tested, e.g., by conducting experiments on synthetic datasets).
- The experiments lack cross-validation – or why else would error bars/confidence intervals not be applicable in your setting (see Checklist Item 3(c))?

As a consequence, it is unclear
- if PC-Winter outperforms Data Shapley in other, equally relevant settings (notably, Data Shapley also outperforms PC-Winter in some settings, which is only conceded in the appendix),
- to which extent the valuations derived are consistent across settings (which is crucial for valuation when the downstream usage of the graph data evaluated is not known precisely), and
- if the experimental results presented are robust.

**Documentation:**

The authors make the code available on anonymous GitHub (I would encourage them to transition to the regular GitHub once their manuscript gets accepted – plus, anonymization is not even required on this track and the authors do not anonymize their manuscript either), but the paper could benefit from a pointer to the code in the main text (e.g., at the beginning of the Experiments section). Also, please include a license for your code (see Checklist Item 4(b)).

Although the appendix describes the experimental setup in more detail, there is still a lot of room for clarification (see "Clarity").

**Ethics:**

While I do have concerns regarding the dual-use potential of the proposed method that are relevant for ethics flag 4 ("Safety and security"), I believe that the authors can address these concerns by including a well thought-out impact statement in the main part of their paper (see also "Limitations").

**Limitations:**

The paper currently lacks a discussion of its potential negative societal impact (both in the main paper and in the appendix), and the discussion of limitations is barely existent (mostly limited to "further efficiency enhancements are crucial" in l. 432).

I encourage the authors to think through, and elaborate on, the potential negative implications of their work, especially with regard to its potential usage to inform data poisoning or other adversarial attacks (on the flipside, you could also discuss other applications, e.g., in data cleaning or explainability, to broaden the relevance of your work).

Furthermore, I encourage the authors expand their discussion of limitations and avenues for future work, e.g., to include the need to understand better the scenarios in which PC-Winter does not outperform Data Shapley (some of which you show, e.g., on page 17 in your appendix), to extend experiments beyond inductive node classification and beyond a single GNN variant, to provide data valuation for graph-learning models other than GNNs, and to make sense of the different "change patterns" occurring when you run the same experiment on different datasets (unless, of course, these limitations are addressed in a revised set of experiments, see also "Correctness").

**Opportunities For Improvement:**

See above

**Relation To Prior Work:**

The discussion of related work in the main text is limited, but a more extensive related work section is provided in the appendix.  However, while the description of the game-theoretic toolkit and related work appears comprehensive, the part on Graph Neural Networks lacks nuance (to the point of not being helpful at all), both in the main text and in the appendix.

**Summary And Contributions:**

The paper addresses the graph-data valuation problem, proposing the Precedence-Constrained Winter (PC-Winter) Value to quantify the utility of nodes and edges for graph-learning tasks and introducing approximations for its computation.

---

> ### Author Rebuttal · Authors · 2024-08-19
>
> **Q1."The paper currently lacks a discussion of its potential negative societal impact (both in the main paper and in the appendix), and the discussion of limitations is barely existent (mostly limited to "further efficiency enhancements are crucial" in l. 432)."**
>
> A1: We appreciate the reviewer's observation regarding the lack of discussion on potential negative societal impacts and limitations. We acknowledge this oversight and will address it in the revised manuscript. Here's an expanded discussion:
> Potential Negative Societal Impacts:
> 1. Privacy concerns: PC-Winter's ability to identify high-value nodes could potentially be misused to target influential individuals in social networks, raising privacy concerns.
> 2. Adversarial attacks: Knowledge of high-value nodes or edges could be exploited to design more effective adversarial attacks on graph-based systems.
> 3. Overreliance on automated valuation: Uncritical use of PC-Winter for decision-making in sensitive domains could lead to overlooking important qualitative factors not captured by the method.
>
> ---
>  **Q2."I encourage the authors expand their discussion of limitations and avenues for future work, e.g., to include the need to understand better the scenarios in which PC-Winter does not outperform Data Shapley (some of which you show, e.g., on page 17 in your appendix), to extend experiments beyond inductive node classification and beyond a single GNN variant, to provide data valuation for graph-learning models other than GNNs, and to make sense of the different "change patterns" occurring when you run the same experiment on different datasets"**
>
> A2:
> We appreciate the reviewer's suggestion to expand our discussion of limitations and future work. We would like to discuss these points as follows.
>
> **Scenarios where PC-Winter doesn't outperform Data Shapley:**
> We thank the reviewer for noting the similarity in performance between PC-Winter and Data Shapley in the Mixed Node Dropping Experiment (Appendix E, Figure 6). We would like to discuss these experiments further and clarify their relationship to other experiments in the main paper (Figure 3) and Appendix F (Figure 7). Together, these experiments provide a comprehensive evaluation of PC-Winter and other baselines, including Data Shapley. As discussed in Section 4.1, "Labeled nodes often contribute more significantly to model performance than unlabeled nodes because they directly offer supervision." Therefore, with accurately assigned node values, labeled nodes should be prioritized for removal over unlabeled nodes. The experiments in Figure 6 (Appendix E) empirically validate this hypothesis, which we explicitly discuss in Section 4.1 (Lines 339-348).
>
> The results in Figure 6 (Appendix E) demonstrate that the majority of labeled nodes are removed prior to the unlabeled nodes by both PC-Winter and Data Shapley. This leads to a plateau in the latter portion of the performance curves since a GNN model cannot be effectively trained with only unlabeled nodes. Such a scenario significantly hampers the ability to assess the value of unlabeled nodes. This is why we proposed separately assessing the values of labeled nodes and unlabeled nodes. Figure 3 in the main paper assesses the quality of values for unlabeled nodes, which are typically smaller and more challenging to estimate accurately. In contrast, Figure 7 in Appendix F focuses on the values of labeled nodes.
>
> We would like to clarify that the experiments in Figure 6 (Appendix E), Figure 7 (Appendix F), and Figure 3 (main paper) are not isolated; rather, they form a coherent series with logical progression and interdependencies. Overall, our method, PC-Winter, demonstrates strong performance in accurately assessing the value of both labeled and unlabeled nodes, particularly in addressing the challenges associated with the latter.
>
> We will revise our manuscript to make these points clearer.
>
> **Extending beyond inductive node classification:**
> As detailed in Appendix G.1, we mainly focus on the inductive node classification setting in this paper. In real-world scenarios, graphs are often large, and the available labels are often sparsely distributed. The inductive setting better reflects conditions where training, validation, and test nodes are distant and do not directly influence each other. This is why the inductive setting, which aims to generalize a trained model to unseen nodes, is widely adopted in real-world graph applications [1, 2, 3, 4]. Unlike the transductive setting, the inductive setting separates test nodes from the training graph, allowing us to measure the value of graph elements solely based on their contribution to GNN model training, without the potential bias introduced by proximity to test or validation nodes.
>
> On the other hand, in the transductive setting, the validation and test nodes have connections to the training nodes. Hence, the training nodes may contribute to both the training process and inference process. This could lead to overvaluation of certain nodes due to their proximity to validation/test nodes. Therefore, data valuation in the transductive setting faces distinct challenges compared to the inductive setting. While it is worthwhile to test PC-Winter in the transductive setting, we recognize that PC-Winter is not specifically designed to address the unique challenges posed in the transductive setting. As such, we did not conduct specific tests in the transductive context in this work, but we plan to explore this in future research.
>
> In summary, the inductive setting is a practical and commonly used setting in real-world graph applications. Furthermore, the inductive and transductive settings each present distinct challenges. PC-Winter is designed to tackle the challenges in the inductive setting. Hence, in this paper, we mainly conduct experiments in the inductive setting.
>
> ...

---

> > ### Author Rebuttal · Authors · 2024-08-19
> >
> > ...
> >
> > **Analyzing pattern differences across datasets:**
> > We agree that a deeper analysis of the different "change patterns" observed across datasets could provide valuable insights. This investigation could reveal how specific graph properties influence valuation effectiveness and guide practitioners in choosing appropriate valuation methods for their data. While this is an interesting direction, we believe it is beyond the scope of the current paper. We plan to explore this in future work by generating synthetic graphs with controlled properties (e.g., degree distribution, clustering coefficient, community structure) to systematically investigate how these properties affect valuation patterns.
> >
> > **Limited GNN variant evaluation:**
> > Regarding the evaluation of the proposed values' effectiveness on other GNN variants, we kindly direct you to the results presented in A.3 in the subsequent sections of this response.
> >
> > [1] Will Hamilton, Zhitao Ying, and Jure Leskovec. "Inductive representation learning on large graphs"
> > [2] Rafaël Van Belle, Charles Van Damme, Hendrik Tytgat, and Jochen De Weerdt. "Inductive graph representation learning for fraud detection"
> > [3] Theis E Jendal, Matteo Lissandrini, Peter Dolog, and Katja Hose. "Simple and powerful architecture for inductive recommendation using knowledge graph convolutions"
> > [4] Edoardo D'Amico, Khalil Muhammad, Elias Tragos, Barry Smyth, Neil Hurley, and Aonghus Lawlor. "Item graph convolution collaborative filtering for inductive recommendations"
> >
> > ---
> >
> > **Q3. "The evaluation is conducted using only one GNN variant (please also provide a high-level description of the architecture in the main text and a detailed specification in the appendix)."**
> >
> > A3: We appreciate the reviewer's observation about the limited GNN variant evaluation. To address this, we have expanded our experiments to include APPNP, another popular GNN variant, alongside the GCN architecture used in our initial experiments. We conducted additional experiments using both GCN and APPNP as downstream models for node classification tasks on the Cora, Citeseer, and Pubmed datasets. The results are presented in a new figure in the attached PDF. We focus on comparing PC-Winter with Data Shapley as it was the strongest baseline based on our experiments in Figures 3 and 4 of the original paper. Analysis of the results shows that PC-Winter consistently outperforms Data Shapley across both GNN architectures. For GCN, PC-Winter achieves lower accuracy curves on all datasets, indicating better identification of important nodes. With APPNP, PC-Winter maintains its advantage, particularly on Citeseer and Pubmed, where the gap in performance is relatively large. The consistent superior performance of PC-Winter across different GNN architectures demonstrates its robustness and effectiveness in graph data valuation tasks, further validating its capability in identifying high-value graph elements, regardless of the specific GNN architecture used for the downstream task.
> >
> > ---
> > **Q4."The evaluation tasks are limited to experiments on inductive node classification (despite the abstract promising "diverse datasets and tasks")."**
> >
> > A4:  We appreciate the reviewer's observation regarding the scope of our evaluation tasks. We'd like to clarify and expand on this point:
> >
> > Task Diversity: While our primary evaluation metric is indeed inductive node classification, our experiments encompass two distinct tasks that provide different perspectives on data valuation:
> >
> > a) Node Dropping: This task evaluates the impact of removing high-value nodes, simulating scenarios where we need to identify critical data points.
> > b) Edge Addition: This task assesses the effect of adding high-value edges, mimicking situations where we aim to enhance the graph structure most efficiently.
> >
> > These two tasks, although based on the same underlying classification problem, offer diverse insights into the utility of our method for different graph modification scenarios.
> >
> > Inductive Setting Rationale: Regarding the inductive setting, we kindly direct you to the discussion we presented at A2 above, "Extending beyond inductive node classification" part.
> >
> > ---
> >
> > **Q5."It is unclear how robust the data valuations are to noise in the data (this could be tested, e.g., by conducting experiments on synthetic datasets)."**
> >
> > A5: We appreciate the reviewer's important question about the robustness of our data valuations to noise. Our response addresses this concern from three perspectives:
> >
> > **Inherent Data Noise Identification in Data Valuation**
> > While not specifically designed for noise identification, data valuation methods including PC-Winter and Data Shapley can naturally be applied to identify low-quality or noisy data. In fact, all the datasets we used in our experiments contain some inherent noise. In our node-dropping experiments (Figure 3), we observed that the performance curves of PC-Winter and Data Shapley eventually rebound toward the end in all datasets. As we analyzed in Section 4.1, this rebound corresponds to the removal of unlabeled nodes that make negative contributions to model performance. These "negative" unlabeled nodes at the end of the curves are indicative of low-quality or noisy data, as their removal helps improve model performance. In Figure 3, the pattern of a performance decrease at the beginning followed by an increase toward the end of the curves suggests that both PC-Winter and Data Shapley can effectively differentiate between high-quality and low-quality (noisy) nodes. Notably, PC-Winter tends to reach a deeper low point than Data Shapley and also experiences a steeper increase at the end. This demonstrates that PC-Winter is more effective than Data Shapley in distinguishing high-quality nodes from noisy ones.
> >
> > In summary, both PC-Winter and Data Shapley are capable of identifying low-quality or noisy data, with PC-Winter outperforming Data Shapley in this regard.
> >
> > ...

---

> > > ### Author Rebuttal · Authors · 2024-08-19
> > >
> > > ...
> > >
> > > **Label Noise Experiment**
> > > We further validate PC-Winter's robustness to label noise. We conducted an experiment on intentionally corrupted datasets. We randomly altered the labels of 40% of nodes in the Cora and Citeseer datasets. Among the top 10 nodes ranked by PC-Winter, 8 were indeed mislabeled in both datasets. Of the 20 nodes with the lowest PC-Winter values, 14 in Cora and 15 in Citeseer were mislabeled. This demonstrates PC-Winter's strong ability to identify mislabeled nodes effectively, further confirming its robustness to label noise.
> > >
> > > **Utility Function Noise Experiment**
> > > To address the more general concept of data valuation robustness, we drew inspiration from [1]. They introduced the idea of evaluating data valuation methods against noisy utility functions. Following this concept, we designed a robustness test. Specifically, we introduced controlled noise to the utility function by multiplying its output by (1 + ε), where ε is uniformly sampled from [-0.01, 0.01], simulating up to ±1% random perturbation in the performance values. Note that such a random perturbation is reasonable and practical, as the standard deviation of multiple runs of GNN models is often close to 1% of their model performance [2]. We generated 100 permutations for smaller datasets and 30 for larger ones, computed both clean and noisy PC-Winter and Data Shapley values, and then, following [1], calculated Spearman rank correlation coefficients to quantify the impact of noise on value rankings. The Spearman rank correlation coefficient measures the monotonic relationship between two rankings. In our context, a coefficient close to 1 indicates that the relative ordering of node values is largely preserved despite the introduced noise, suggesting high robustness of the valuation method. Conversely, a coefficient close to 0 implies that the noise disrupts the ranking, indicating low robustness.
> > >
> > > The results, averaged over 10 random seeds, are as follows:
> > > | Dataset   | PC-Winter Correlation | Data Shapley Correlation |
> > > |-----------|----------------------:|-------------------------:|
> > > | Cora      |               0.9465 |                  0.7502 |
> > > | Citeseer  |               0.9511 |                  0.7204 |
> > > | Pubmed    |               0.7202 |                  0.2372 |
> > > | Photo     |               0.4950 |                  0.3076 |
> > > | Computers |               0.3169 |                  0.1888 |
> > > | Physics   |               0.3882 |                  0.0872 |
> > >
> > > The results demonstrate that PC-Winter consistently exhibits surprisingly stronger robustness than Data Shapley across all datasets. These results provide compelling evidence of PC-Winter's robustness to noise in the utility function. We acknowledge that it is important to dedicate future efforts to conducting a full investigation into the underlying reasons for PC-Winter's robustness. However, investigating robustness in the presence of noise in the utility function, as discussed in studies such as [1], would require significant effort and is beyond the scope of this paper. Therefore, we plan to address this in future work.
> > >
> > > In conclusion, our experiments provide strong evidence of PC-Winter's robustness to various types of noise: data noise which may come from structure or features (low-value nodes), label noise, and utility function noise. We would like to further investigate the robustness of PC-Winter from different perspectives in future work.
> > >
> > > [1] Wang, Jiachen T., and Ruoxi Jia. "Data banzhaf: A robust data valuation framework for machine learning."
> > > [2] Shchur, Oleksandr, et al. "Pitfalls of graph neural network evaluation."
> > >
> > > ---
> > >
> > > **Q6."The experiments lack cross-validation – or why else would error bars/confidence intervals not be applicable in your setting (see Checklist Item 3(c))?"**
> > >
> > > A6:
> > > Thank you for the question regarding cross-validation and error bars. We would like to clarify why these are not directly applicable in our experimental setting.
> > >
> > > Our experimental setup follows established practices in data valuation literature [1-5], including recent works on Data Shapley and its variants. These studies evaluate data valuation methods through deterministic element removal or addition processes, where error bars and confidence intervals are not applicable. We follow a similar setup to design the node dropping and edge addition experiments for evaluating graph data valuation methods. Hence, error bars and confidence intervals are not applicable either in our case. In particular, we would clarify the following points:
> > >
> > > **Fixed sequence after valuation**: Once PC-Winter values are estimated for graph elements, the sequence of node-dropping or edge addition operations becomes deterministic. This fixed ordering means that repeating the experiment with the same valuation would yield identical results.
> > >
> > > **Nature of the evaluation task**: Our experiments aim to evaluate how well the estimated values capture the importance of graph elements. This is fundamentally different from assessing model performance variability, where cross-validation and error bars are typically used.
> > >
> > > We thank the reviewer for raising this important point, which helps clarify the nature of our evaluation methodology. We will revise the manuscript to clarify this point.
> > >
> > > [1] Ghorbani, Amirata, and James Zou. "Data shapley: Equitable valuation of data for machine learning."
> > > [2] Kwon, Yongchan, and James Zou. "Beta shapley: a unified and noise-reduced data valuation framework for machine learning."
> > > [3] Wang, Jiachen T., and Ruoxi Jia. "Data banzhaf: A robust data valuation framework for machine learning."
> > > [4] Yoon, Jinsung, Sercan Arik, and Tomas Pfister. "Data valuation using reinforcement learning."
> > > [5] Jiang, Kevin, et al. "Opendataval: a unified benchmark for data valuation."
> > >
> > > ---

---

> > > > ### Author Rebuttal · Authors · 2024-08-19
> > > >
> > > > **Q7."We utilize G_{tr} to train the GNN model, which is evaluated on V_{va} (G_{va}?) for obtaining the data values for elements. The test graph V_{te} (G_{te}?) is utilized to evaluate the effectiveness of the obtained values. => Can you clarify what this means? A small drawing depicting your experimental process on a toy graph might be particularly helpful."**
> > > >
> > > > A7:
> > > > We appreciate the reviewer's request for clarification on our experimental setup. We acknowledge that our description could be clearer, and we're happy to provide more details. Our approach is an extension of established data valuation pipelines [1] to graph-structured data. Here's a step-by-step explanation of our process:
> > > >
> > > > **Initial graph partitioning:**
> > > > Given a large input graph, we first identify training, validation, and testing nodes.
> > > > We construct the training graph $G_{tr}$ by including all training nodes and their 2-hop neighbors, excluding any validation or testing nodes. We create separate subgraphs $G_{va}$ and $G_{te}$ induced by the validation and testing nodes, respectively.
> > > >
> > > > **Model training and data valuation:**
> > > > We train the GNN model on $G_{tr}$.
> > > > To compute data values, we use $G_{va}$ as part of the utility function. Specifically, the model's accuracy on $G_{va}$ serves as the performance metric for data valuation.
> > > >
> > > > **Value-based graph modification:**
> > > > Based on the estimated data values, we modify $G_{tr}$ by removing nodes or adding edges.
> > > > We then retrain the GNN model on this modified training graph.
> > > >
> > > > **Evaluation:**
> > > > Finally, we evaluate the retrained model's performance on $G_{te}$ to assess the effectiveness of our data valuation method.
> > > >
> > > > This setup allows us to simulate realistic scenarios where data valuation informs graph modifications, and the impact is measured on a held-out set. We thank the reviewer for highlighting this need for clarity, which will undoubtedly improve the accessibility of our work. In the revision, we will try to create a figure to illustrate this process to make it clearer.
> > > >
> > > > [1] Jiang, Kevin, et al. "Opendataval: a unified benchmark for data valuation."
> > > >
> > > > ---
> > > >
> > > > **Q8."However, while the description of the game-theoretic toolkit and related work appears comprehensive, the part on Graph Neural Networks lacks nuance (to the point of not being helpful at all), both in the main text and in the appendix."**
> > > >
> > > > A8: We appreciate the reviewer's feedback on our GNN discussion.
> > > >
> > > > Our primary focus in this paper is on Graph Data Valuation, which aims to measure the utility of nodes and edges in graph datasets. Our work leverages the general principles of GNNs, particularly their feature propagation mechanism and the concept of the computation tree, to develop more effective graph data valuation techniques. This is why we dedicate effort to explain these concepts in Section 2.3 in the main text. We believe this preliminary knowledge is essential for readers to fully understand the methodology (PC-Winter) proposed in Section 3.
> > > >
> > > > That being said, we agree that providing more context on GNNs could enhance the reader's understanding of our method. In our revision, we plan to: 1) Expand the discussion on how GNN principles, especially feature propagation and computation graphs, inform and enable our graph data valuation approach; 2) Provide a brief overview of common GNN architectures (e.g., GCN, APPNP, and SGC) and explain why we chose our specific architecture for this work; 3) In Appendix A.2, offer a more comprehensive review of GNNs, their variants, and applications.
> > > >
> > > > We believe these additions will provide readers with sufficient context to understand the GNN aspects of our work without detracting from our main contribution in graph data valuation.
> > > >
> > > > ---
> > > >
> > > > **Q9."The authors make the code available on anonymous GitHub (I would encourage them to transition to the regular GitHub once their manuscript gets accepted – plus, anonymization is not even required on this track and the authors do not anonymize their manuscript either), but the paper could benefit from a pointer to the code in the main text (e.g., at the beginning of the Experiments section). Also, please include a license for your code (see Checklist Item 4(b))."**
> > > >
> > > > A9. Thank you for the suggestion. We will improve the code documentation by adding more detailed comments and explanations. We will enhance the code documentation by adding more detailed comments and explanations. We are working on it and will provide an updated version at [our GitHub repository](https://github.com/frankhlchi/graph-data-valuation).
> > > >
> > > > ---
> > > >
> > > > **Q10."Observation 1 (ll. 133–140) seems redundant, given the challenges you state in the introduction."..."l. 222 and Eq. 3 (probably also in other places): There seem to be curly braces missing around p in "D(p) \cup {p}"."**
> > > >
> > > > A10: Thank you for bringing these important points to our attention. We appreciate your thorough review and helpful suggestions. We will address each of these issues in our revised manuscript.

---

> > ### Comment · Reviewer_deQC · 2024-08-25
> > **Response to Rebuttal Comments**
> >
> > Thank you for answering the questions and concerns raised by all reviewers in detail.
> > I appreciate that you invested significant effort to improve your work based on reviewers' feedback and will recognize this by raising my score.
> >
> > Two additional comments:
> > 1. Regarding the **inductive node-classification setting**: I know that this is a standard setting, and I know that your method is tailored to that setting. What I take issue with is your trying to "oversell" PC-Winter as a general method _although_ it is a method specifically developed for the inductive node-classification setting (and, hence, not only excluding the transductive setting but also other [non-node-focused] tasks). So please make sure that you clearly specify the scope of your work.
> > 2. Regarding the **error bars**: I understand that error bars aren't applicable in your specific setup because the processes you study are deterministic, and that this is common in the data-valuation literature (which, however, may not yet have converged on _best_ experimental practices). However, given the questions around noise and robustness in other parts of the review and rebuttal, it might be prudent to consider exactly the same experiment but with randomized node removal, e.g., in the sense that you don't always remove the top element but rather one of the "top k" or all "within-distance-eps-of-the-top" elements drawn randomly, for varying values of k and varying seeds – which then would allow you to report error bars, too. The "hunch" here is that the ordering of nodes determined at the start is probably less of a linear and more of a partial order (what is the distribution of differences between data valuations you observe when creating that ordering, by the way?), and a good valuation method should perform consistently when treating such small fluctuations as irrelevant to the ordering. Can you comment on that possibility?

---

> > > ### Author Response · Authors · 2024-08-25
> > >
> > > Thank you for your reply and additional insightful comments and suggestions. We greatly appreciate your continued engagement and the valuable insights you've provided to help improve our work.
> > >
> > > To further address your additional points:
> > >
> > > (1) Regarding the inductive node-classification setting, we agree that it's crucial to clearly define the scope of our work. In response, we will add a detailed explanation in the appendix about our rationale for choosing the inductive node-classification setting and clarify that our current trained model focuses specifically on node classification tasks.
> > >
> > > (2) Concerning the error bars, we find your suggestion both innovative and valuable, not just for our work but for the entire field of data valuation evaluation. Based on our observations, the node value distribution exhibits a pronounced peak near zero with a noticeable right-skew and a slightly heavy right tail. While most values concentrate around zero, there are occasional large positive values and a few negative outliers. Regarding your suggestion, we think it is possible to extend your idea using importance sampling, which offers a more nuanced approach than simply selecting from the top k elements. Specifically, we can use the node values as softmax scores (potentially with a temperature parameter) to create an importance sampling distribution for node removal or edge addition. By conducting multiple sampling experiments using this distribution, we can estimate error bars and provide a more robust data valuation method evaluation paradigm.

---

> > > > ### Comment · Reviewer_deQC · 2024-08-30
> > > > **Response to Additional Author Comments**
> > > >
> > > > Thank you for reacting to my additional comments.
> > > > I appreciate your responsiveness to reviewer feedback and believe that the additional changes you suggested will further strengthen the manuscript.
> > > > With better-hedged claims and a more robust evaluation, your submission would be a valuable addition to the data-centric work also in scope of the NeurIPS Datasets and Benchmarks track, and consequently, I will support its publication.

---

> > > > > ### Author Response · Authors · 2024-08-30
> > > > >
> > > > > Thank you for your continued engagement and guidance. We appreciate your thorough review and support. Your insights have been invaluable in improving our work.

---

### Official Review · Reviewer_C9gV · 2024-07-25
**More of a research track article**

**Rating:** 6
**Confidence:** 4

**Review:**

This Shapley-meets-graph work is significant as it advances the field of graph data analysis by providing a method to quantify the value of data in a way that is both meaningful and computationally feasible. This has important implications for various applications, such as  enhancing data efficiency, improving model interpretability, and guiding data collection strategies.

The emphasis of the article extends beyond just datasets and benchmarks, and in my opinion, is more suitable for an attack article on GNNs with Shapley-based valuation methods. This could lead to more nuanced insights into data contribution within graphs.  As it stands, I am not sure how the insights in figure 3 (which are quite interesting) helps us evaluate datasets. Can something be quantified about the information shown in the figures?  The "benchmarking datasets" aspect  is interesting but needs a clear goal and output measure.

**Strengths:**

- PC-Winter Value introduces a novel way to apply cooperative game theory in graph data settings, expanding the applicability of Shapley values.
- The authors conduct tests on various datasets to validate the effectiveness and robustness of the proposed method.
- The approach has the potential to significantly impact how researchers and practitioners evaluate and utilize graph data in machine learning models.
-  The method is applicable in various real-world scenarios where understanding the contribution of data points within a graph is crucial.

**Additional Feedback:**

- Are there specific types of graph data where PC-Winter might not perform as well?
- How does the complexity of the graph data (e.g., density of edges) affect the computation and accuracy of PC-Winter values?

**Clarity:**

The submission is clear and well-structured, making it accessible to readers familiar with graph neural networks and data valuation methods. The paper clearly defines the problem, the proposed solution, and the results of the experiments. However, the provided code lacks comments, making it less accessible for others to learn from or build upon. Improving documentation would help other researchers.

**Correctness:**

- The evaluation methods chosen are appropriate for the type of research being conducted. The use of various real-world datasets ensures that the results are generalizable across different types of graph data.
- The datasets used (Cora, Citeseer, Pubmed, and others) are standard in graph machine learning, which ensures that the results are relevant to ongoing research in the field.

**Documentation:**

- The details about how graph datasets are organized and used in experiments are standard in the field, and the paper likely adheres to these norms, using nodes and edges with features and labels appropriate for GNN tasks.
- The paper does not deal with a new dataset directly, but it uses existing datasets to support its claims and experimental validation. For benchmarks, the paper provides sufficient details to ensure that the experiments are reproducible, though improvements could be made in the accessibility and documentation of the experimental code.

**Limitations:**

Not the main article but line 576 mentions a single limitation, but I fond the text inadequate for a Neurips submission.

**Opportunities For Improvement:**

- While the article mentions computational challenges, it could benefit from a deeper discussion on the limitations and potential biases of the proposed valuation method.
- A clear output that shows the challenge of training a model on a graph would be a great contribution of this work. Information in figs 3 and 4 should be quantified.

**Relation To Prior Work:**

Related work is lacking - although the authors give all the subfields (2.1-2.3), the main field, i.e., graph data quality, does not have any related work listed.

**Summary And Contributions:**

The article introduces the Precedence-Constrained Winter Value, adapting the Shapley value concept for graph data valuation. This method addresses the challenges of applying traditional data valuation techniques to graph-structured data by accounting for the complex dependencies among graph nodes. The primary contribution of this work is the development of a framework that quantifies the impact of individual nodes and edges on model performance, supported by experiments on various datasets.

---

> ### Author Rebuttal · Authors · 2024-08-17
>
> **Q1.  "The emphasis of the article extends beyond just datasets and benchmarks, and in my opinion, is more suitable for an attack article on GNNs with Shapley-based valuation methods."**
>
> A1.
> We appreciate the reviewer's perspective and would like to clarify our position:
> 1. While the methods we develop can potentially be used for adversarial purposes, the core contribution of our work is introducing graph data valuation, which has many applications beyond attacks.
> 2. Our work aligns well with the Datasets and Benchmarks track.
>
> Firstly, we would like to clarify that our focus in this paper is on Graph Data Valuation, which offers a tool for measuring the utility of nodes and edges in graph datasets. We acknowledge that the node dropping experiments can be regarded as a potential method for attack. However, in our paper, it is primarily a way of evaluating the quality of the obtained data values. Such "data removal" experiments are widely adopted for evaluation in conventional Data Valuation literature for Euclidean data [1-5], and we follow these settings to extend them to the graph data valuation scenario. Furthermore, we would like to emphasize the significance of the Data Valuation problem. Data valuation is a critical and rapidly evolving area in machine learning, with numerous recent works [1-3] highlighting its importance. Its applications span far beyond adversarial scenarios, including facilitating fair compensation [4], guiding efficient data collection [5], and improving model performance through informed data selection [6]. Our work takes a pioneering position by introducing the graph data valuation problem and proposing an effective solution. In essence, graph data valuation is a novel and important problem in its own right, with potential impacts comparable to those of traditional data valuation.
>
> Secondly, our work aligns closely with the scope of the Datasets and Benchmarks track. The NeurIPS 2024 call for this track explicitly welcomes "Data-centric AI methods and tools, e.g. to measure and improve data quality or utility, or studies in data-centric AI that bring important new insight." Our method directly addresses the measurement of data utility in graph-structured datasets, a crucial aspect of data-centric AI. By quantifying the value of nodes and edges, we provide a tool for assessing and potentially improving the quality of graph datasets, which is aligned with the goals of this track.
>
> [1] Yoon, Jinsung, Sercan Arik, and Tomas Pfister. "Data valuation using reinforcement learning."
> [2] Ghorbani, Amirata, Michael Kim, and James Zou. "A distributional framework for data valuation."
> [3] Wang, Jiachen T., and Ruoxi Jia. "Data banzhaf: A robust data valuation framework for machine learning."
> [4] Ghorbani, Amirata, and James Zou. "Data shapley: Equitable valuation of data for machine learning."
> [5] Jiang, Kevin, et al. "Opendataval: a unified benchmark for data valuation.
> [6] Ghorbani, Amirata, James Zou, and Andre Esteva. "Data shapley valuation for efficient batch active learning."
>
> ---
>
> **Q2.  "As it stands, I am not sure how the insights in figure 3 (which are quite interesting) helps us evaluate datasets. Can something be quantified about the information shown in the figures? The "benchmarking datasets" aspect is interesting but needs a clear goal and output measure."**
>
> A2. We appreciate the reviewer's suggestion to quantify the insights from Figures 3 and 4. To address this, we have calculated two key metrics for each method across all datasets:
>
> 1. For the node dropping experiment (Figure 3):
>    a) Minimum Accuracy: The lowest accuracy reached during node removal.
>    b) Mean Accuracy: The average accuracy across all node removal steps.
>
> 2. For the edge addition experiment (Figure 4):
>    a) Maximum Accuracy: The highest accuracy reached during edge addition.
>    b) Mean Accuracy: The average accuracy across all edge addition steps.
>
> The rationale for these metrics is as follows:
>
> For node dropping:
> - Minimum Accuracy: A lower value indicates better identification of the most critical nodes, whose removal causes the largest performance drop.
> - Mean Accuracy: A lower value suggests consistent identification of important nodes throughout the removal process, maintaining lower performance overall.
>
> These metrics capture our goal of finding a method that causes the accuracy curve to drop deeply (Minimum Accuracy) and stay lower for a substantial portion of the removal process (Mean Accuracy).
>
> For edge addition:
> - Maximum Accuracy: A higher value indicates better identification of the most valuable edges, whose addition leads to the largest performance improvement.
> - Mean Accuracy: A higher value suggests consistent identification of important edges throughout the addition process, maintaining higher performance overall.
>
> These metrics reflect our aim to find a method that raises the accuracy curve quickly (Maximum Accuracy) and maintains higher performance throughout the edge addition process (Mean Accuracy). The quantitative results are presented in Tables 1-4 in the attached PDF. This provides concrete measures of PC-Winter's effectiveness in identifying high-value graph elements.
>
> ---

---

> > ### Author Rebuttal · Authors · 2024-08-17
> >
> > **Q3. "While the article mentions computational challenges, it could benefit from a deeper discussion on the limitations and potential biases of the proposed valuation method."**
> >
> > A3: We appreciate the reviewer's suggestion to provide a deeper discussion on the limitations and potential biases of PC-Winter. We agree that this is an important aspect to address, and we will expand our discussion in the paper. Here are the key points we will elaborate on:
> >
> > 1. Truncation bias: As noted, our hierarchical truncation strategy may undervalue 2-hop neighbors and beyond. This bias is a trade-off we make for computational efficiency, but it could lead to underestimation of some nodes' contributions.
> >
> > 2. Permutation sensitivity: Like other permutation-based methods, PC-Winter's accuracy depends on the number of permutations sampled. With fewer permutations, marginal contributions may be over- or underestimated due to positional effects in the sampled permutations.
> >
> > 3. Vulnerability to adversarial manipulation: High-value neighbors identified by PC-Winter could potentially be targeted for adversarial attacks, as they are recognized as influential to the model's performance. This vulnerability arises from the very nature of what PC-Winter aims to achieve - identifying the most impactful nodes and edges in a graph. Specifically, an adversary with knowledge of PC-Winter's valuation could focus their efforts on manipulating or corrupting the highest-valued nodes or edges. This targeted approach could be more efficient and damaging than random attacks.
> >
> > ---
> > **Q4. "However, the provided code lacks comments, making it less accessible for others to learn from or build upon. Improving documentation would help other researchers."**
> >
> > A4. Thank you for your feedback. We will enhance the code documentation by adding more detailed comments and explanations. We are working on it and will provide an updated version at [our GitHub repository](https://github.com/frankhlchi/graph-data-valuation).
> >
> > ---
> >
> > **Q5. "Related work is lacking - although the authors give all the subfields (2.1-2.3), the main field, i.e., graph data quality, does not have any related work listed."**
> >
> > A5:  We appreciate the reviewer's insightful comment on related work. We agree that a comprehensive discussion of relevant fields is crucial for contextualizing our research.
> >
> > Our work is primarily situated within the field of data valuation, with a specific focus on graph-structured data. Data valuation is indeed a well-established and rapidly growing research area, as evidenced by seminal works such as Data Shapley [1] and subsequent developments like Beta Shapley [2] and Data Banzhaf [3].
> >
> > This is why we have dedicated substantial effort to introducing related work in Data Valuation, both in the main paper and the appendix. Our comprehensive review of this field serves to contextualize our work within the broader landscape of data valuation research and highlight the unique challenges posed by graph-structured data. Furthermore, we have also placed significant emphasis on discussing related work in Shapley values on graph data, as this represents the intersection of data valuation techniques and graph-structured data – a key area for our work. This discussion can be found in Section A.3 of our appendix, where we provide a detailed overview of how Shapley values have been applied to graph neural networks, particularly in the context of explainability and feature importance.
> >
> > We thank the reviewer for bringing attention to the field of graph data quality. While our primary focus is on data valuation for graphs, we recognize that graph data quality is indeed a relevant and important adjacent field. We appreciate this suggestion and agree that our paper would benefit from a more detailed discussion of graph data quality. In our revision, we will expand our related work section to include key references and concepts from graph data quality, highlighting how our work on graph data valuation complements and potentially contributes to this field.
> >
> > [1] Ghorbani, Amirata, and James Zou. "Data shapley: Equitable valuation of data for machine learning."
> > [2] Kwon, Yongchan, and James Zou. "Beta shapley: a unified and noise-reduced data valuation framework for machine learning."
> > [3] Wang, Jiachen T., and Ruoxi Jia. "Data banzhaf: A robust data valuation framework for machine learning."
> >
> > ---
> >
> > **Q6. "Are there specific types of graph data where PC-Winter might not perform as well?"**
> >
> > A6: We appreciate this insightful question about PC-Winter's performance across different graph types. While our current experiments focus on homogeneous graphs, we can offer some informed inference about its potential performance on other graph types. Specifically, for heterogeneous graphs, PC-Winter was originally designed with homogeneous graphs in mind. When applied to heterogeneous graphs, some information about node and edge types would need to be omitted or simplified, which could potentially impact performance. The method might not fully capture the complex interactions between different node and edge types, leading to less accurate valuations. Those assumptions need more complete work to validate. We hope this work encourages further exploration and validation of PC-Winter's performance on various graph structures.

---

> > ### Author Rebuttal · Authors · 2024-08-17
> >
> > **Q7. "How does the complexity of the graph data (e.g., density of edges) affect the computation and accuracy of PC-Winter values?"**
> >
> > A7.We appreciate the reviewer's insightful question about the impact of graph complexity on PC-Winter. The complexity of the graph, particularly its edge density, indeed affects both the computational cost and the accuracy of PC-Winter values. We have conducted a detailed time complexity analysis in Appendix H.5. For convenience, the analysis is conducted for a d-regular graph, where each node has a degree $d$. In the analysis, the time complexity of PC-Winter for a single permutation is $O(L · N_{trun}· (N_{trun}/2 · F + F^2))$, where $N_{trun} = 1 + d \cdot (1-r_1) + d^2 \cdot (1-r_1)(1-r_2)$ with $r_1$, $r_2$ denoting the truncation ratios, $F$ represents the dimensionality of node features in each layer of the GNN model, and L is the number of labeled nodes. Clearly, the time complexity grows with $d$, indicating that graph density influences computational cost. Furthermore, this could indirectly affect the quality of estimated data values. In particular, under the same time constraints, denser graphs allow for fewer permutations to be processed, which could potentially impact the accuracy of the estimated values. However, our hierarchical truncation strategy effectively mitigates this impact to some extent. By setting appropriate truncation ratios, we can keep $N_{trun}$ much smaller than the full computation tree size in dense graphs, thereby improving efficiency.

---

> > > ### Comment · Reviewer_C9gV · 2024-08-23
> > > **thanks**
> > >
> > > Thanks for the responses. I will keep my accept score.

---

> > ### Author Response · Authors · 2024-08-24
> >
> > Thank you for your response! We greatly appreciate your prompt response and the time you have taken to read through our replies. We believe that our responses have addressed the questions and concerns you raised in your initial review. However, if there are any outstanding issues or further clarifications needed, please do not hesitate to let us know. Given the additional insights provided in our responses, we respectfully request your consideration for an increase in the score of our paper.

---

### Author Response · Authors · 2024-08-23
**GitHub Update**

We appreciate all reviewers for their valuable feedback. In response to your comments:

1. Additional code comments have been added throughout the [repository](https://github.com/frankhlchi/graph-data-valuation) to improve clarity and understanding.

2. The README file has been expanded to include:
   - A visual explanation of the inductive graph split
   - A visual overview of the PC-Winter value estimation procedure

These updates aim to make our work more accessible.

We welcome any further feedback on our rebuttal. Thank you for your time and insights in helping us improve this work.

---

### Decision · Program_Chairs · 2024-09-26

**Decision:**

Reject

**Comment:**

The paper addresses the graph-data valuation problem, proposing the Precedence-Constrained Winter Value to quantify the utility of nodes and edges for graph-learning tasks and introducing approximations for its computation. Reviewers agreed that the paper was well written. They appreciated the novel way and principled approach with rigor and thought it has the potential to significantly impact how graph data is evaluated and utilized in machine learning models. Technically, there were two main concerns. First, the limitations and potential issues should be more thoroughly discussed. Second, experiments should be more comprehensive to emphasize efficiency issues and cover more GNN variants and tasks.

However, while technically appreciated, in the end, this paper is a research rather than a dataset/benchmark paper. This track is not the right venue to publish this work.